# Tight Bounds for Answering Adaptively Chosen Concentrated Queries

**Emma Rapoport**
Tel Aviv University
emmarapoport@gmail.com

**Edith Cohen**
Google Research
and Tel Aviv University
edith@cohenwang.com

**Uri Stemmer**
Tel Aviv University
and Google Research
u@uri.co.il

## Abstract

Most work on adaptive data analysis assumes that samples in the dataset are independent. When correlations are allowed, even the non-adaptive setting can become intractable, unless some structural constraints are imposed. To address this, Bassily and Freund [2016] introduced the elegant framework of *concentrated queries*, which requires the analyst to restrict itself to queries that are concentrated around their expected value. While this assumption makes the problem trivial in the non-adaptive setting, in the adaptive setting it remains quite challenging. In fact, all known algorithms in this framework support significantly fewer queries than in the independent case: At most $O(n)$ queries for a sample of size $n$, compared to $O(n^2)$ in the independent setting.

In this work, we prove that this utility gap is inherent under the current formulation of the concentrated queries framework, assuming some natural conditions on the algorithm. Additionally, we present a simplified version of the best-known algorithms that match our impossibility result.

## 1  Introduction

Adaptive interaction with data is a central feature of modern analysis pipelines, from scientific exploration to model selection and parameter tuning. However, adaptivity introduces fundamental statistical difficulties, as it creates dependencies between the data and the analysis procedures applied to it, which could quickly lead to overfitting and false discoveries. Motivated by this, following the seminal work of Dwork et al. [2015b], a substantial body of work has established rigorous frameworks for addressing this problem. These works demonstrated that various notions of algorithmic stability, and in particular differential privacy (DP) [Dwork et al., 2006], allow for methods which maintain statistical validity under the adaptive setting. Most of the current work, however, focuses on the case where the underlying data distribution is a *product distribution*, i.e., the samples in the dataset are independent of each other. Much less is understood about the feasibility of accurate adaptive analysis when the data exhibits correlations. In this work, we examine the extent to which accurate adaptive analysis remains possible under minimal structural conditions on the data distribution.

Before presenting our new results, we describe our setting more precisely. Let $\mathcal{X}$ be a data domain. We consider the following game between a data analyst $\mathcal{A}$ and a mechanism $\mathcal{M}$.

1. The analyst $\mathcal{A}$ chooses a distribution $\mathcal{D}$ over tuples in $\mathcal{X}^*$ (under some restrictions).
2. The mechanism $\mathcal{M}$ obtains a sample $S \leftarrow \mathcal{D}$.    % We denote $|S| = n$.
3. For $k$ rounds $j = 1, 2, \ldots, k$:
   (a) The analyst $\mathcal{A}$ chooses a query $q_j : \mathcal{X} \to [0, 1]$, possibly as a function of all previous answers given by $\mathcal{M}$ (under some restrictions).
   (b) The mechanism $\mathcal{M}$ obtains $q_j$ and responds with an answer $a_j \in \mathbb{R}$, which is given to $\mathcal{A}$.

39th Conference on Neural Information Processing Systems (NeurIPS 2025).

Note that the analyst $\mathcal{A}$ is *adaptive* in the sense that it chooses the queries $q_j$ based on previous outputs of $\mathcal{M}$, which in turn depend on the sample $S$. So the queries $q_j$ themselves depend on $S$. If instead the analyst $\mathcal{A}$ were to fix all $k$ queries before the game begins, then these queries would be independent of the dataset $S$. We refer to this variant of the game (where all queries are fixed ahead of time) as the *non-adaptive setting*.

The goal of $\mathcal{M}$ in this game is to produce accurate answers w.r.t. the expectation of the queries over the underlying distribution $\mathcal{D}$. Formally, we say that $\mathcal{M}$ is $(\alpha, \beta)$-*statistically accurate* if for every analyst $\mathcal{A}$, with probability at least $1 - \beta$, for every $j \in [k]$ it holds that $|a_j - q_j(\mathcal{D})| \leq \alpha$, where $q_j(\mathcal{D}) := \mathbb{E}_{T \leftarrow \mathcal{D}}\left[\frac{1}{|T|}\sum_{x \in T} q_j(x)\right]$. As a way of dealing with worst-case analysts, the analyst $\mathcal{A}$ is assumed to be adversarial in that it tries to cause the mechanism to fail. We therefore sometimes think of $\mathcal{A}$ as an *attacker*.

The main question here is:

**Question 1.1.** *What is the maximal number of queries one can accurately answer, $k$, as a function of the sample size $n$, the desired utility parameters $\alpha, \beta$, and the type of restrictions we place on the choice of $\mathcal{D}$ and the queries $q_j$ (in Steps 1 and 3a above)?*

The vast majority of the work on adaptive data analysis (ADA) focuses on the case where $\mathcal{D}$ is restricted to be a *product distribution* over $n$ elements (without restricting the choice of the queries $q_j$). After a decade of research, this setting is relatively well-understood: For constant $\alpha, \beta$, there exist computationally efficient mechanisms that can answer $\Theta(n^2)$ adaptive queries, and no efficient mechanism can answer more than that.[1]

The situation is far less well-understood when correlations in the data are possible. Let us consider the following toy example as a warmup. Suppose that the attacker randomly picks one of the following two distributions:

- $\mathcal{D}_0$ = The distribution that with probability $1/2$ returns the sample $(\frac{1}{2}, \frac{1}{2}, \ldots, \frac{1}{2})$ and w.p. $1/2$ returns the sample $(0, 0, \ldots, 0)$.

- $\mathcal{D}_1$ = The distribution that with probability $1/2$ returns the sample $(\frac{1}{2}, \frac{1}{2}, \ldots, \frac{1}{2})$ and w.p. $1/2$ returns the sample $(1, 1, \ldots, 1)$.

Note that in this scenario, a mechanism holding the sample $(\frac{1}{2}, \frac{1}{2}, \ldots, \frac{1}{2})$ cannot accurately answer the query $q(x) = x$, as the true answer could be either $1/4$ or $3/4$. The takeaway from this toy example is that when correlations in the data are possible, then we must impose additional restrictions on our setting in order to make it feasible. There are two main approaches for this in the literature:

1. **Explicitly limit dependencies within the sample [Kontorovich et al., 2022]:** Intuitively, if we restrict the attacker $\mathcal{A}$ to choose only distributions $\mathcal{D}$ that adhere to certain "limited dependencies" assumptions, then the problem becomes feasible. A downside of this approach is that it is typically tied to a specific measure for limiting dependencies, and it is not clear why one should prefer one measure over another.

2. **Limit the attacker to *concentrated queries* [Bassily and Freund, 2016]:** Notice that the toy example above cannot be solved even in the non-adaptive setting, because the description of the hard query $q(x) = x$ does not depend on the sample $S$. So in a sense, it is "unfair" to attempt solving it in the adaptive setting. In other words, *if something cannot be solved in the non-adaptive setting, how can we hope to solve it in the adaptive setting?* Motivated by this, Bassily and Freund [2016] restricted the attacker $\mathcal{A}$ to queries that in the non-adaptive setting are sharply concentrated around their true mean. Specifically, the attacker is restricted to choose queries $q_j$ such that if we were to sample a fresh dataset $T$ from the underlying distribution $\mathcal{D}$ (where $T$ is independent of the description of $q_j$), then with high probability it holds that the empirical average of $q_j$ on $T$ is close to the true mean of $q_j$ over $\mathcal{D}$. Notice that under this restriction, the problem becomes trivial in the non-adaptive setting, as we could simply answer each query using its exact empirical average. In the adaptive setting, however, the problem is quite challenging.

---

[1]See, e.g., Hardt and Ullman [2014], Dwork et al. [2015b,a], Steinke and Ullman [2015], Bassily et al. [2016], Cummings et al. [2016], Rogers et al. [2016], Feldman and Steinke [2017], Nissim et al. [2018], Feldman and Steinke [2018], Shenfeld and Ligett [2019], Steinke and Zakynthinou [2020], Jung et al. [2020], Shenfeld and Ligett [2023], Blanc [2023], Nissim et al. [2023]

In this work we continue the study of this question for concentrated queries. We aim to characterize the largest number of adaptively-chosen concentrated-queries one can accurately answer (without assuming independence in the data). Formally,

**Definition 1.2** (Concentrated queries). *Let $\mathcal{X}$ be a domain, let $\mathcal{D}$ be a distribution over tuples in $\mathcal{X}^*$, and let $\varepsilon, \gamma \in [0,1]$ be parameters. A query $q : \mathcal{X} \to [0,1]$ is $(\varepsilon, \gamma)$-concentrated w.r.t. $\mathcal{D}$ if*

$$\Pr_{S \sim \mathcal{D}} [|q(S) - q(\mathcal{D})| \geq \varepsilon] \leq \gamma,$$

*where $q(S) = \frac{1}{|S|} \sum_{x \in S} q(x)$ is the empirical average of $q$ on $S$ and $q(\mathcal{D}) = \mathbb{E}_{T \leftarrow \mathcal{D}} [q(T)]$ is the expected value of $q$ over sampling a fresh dataset from $\mathcal{D}$.*

For example, if $\mathcal{D}$ is a *product distribution* over datasets of size $n$, then, by the Hoeffding bound, every query $q : \mathcal{X} \to [0,1]$ is $(\varepsilon, \gamma)$-concentrated for every $\varepsilon \geq \sqrt{\frac{\ln 2}{2n}}$ with $\gamma = 2e^{-2n\varepsilon^2}$. This example motivates the following question:

**Question 1.3.** *How many adaptive queries could we efficiently answer when correlations in the data are allowed, but the analyst is restricted to $(\varepsilon, \gamma)$-concentrated queries for $\varepsilon, \gamma$ that are comparable to what is guaranteed without correlations, say $\varepsilon = \frac{1}{\sqrt{n}}$ and $\gamma = n^{-10}$?*

Bassily and Freund [2016] introduced this question and presented *noise addition mechanisms* that can efficiently answer $O(n)$ adaptive queries under the conditions of Question 1.3. By *noise addition mechanism* we mean a mechanism that given a sample $S$ answers every query $q$ with $q(S) + \eta$, where $\eta$ is drawn independently from a fixed noise distribution. Note the stark contrast from the i.i.d. case, where it is known that $O(n^2)$ queries can be supported rather than only $O(n)$. To achieve their results, Bassily and Freund [2016] introduced a stability notion called *typical stability* and showed that (1) noise addition mechanisms with appropriate noise are typically stable; and (2) typical stability guarantees statistical validity in the adaptive setting, even in the face of correlations in the data. More generally, the algorithm of Bassily and Freund [2016] can support roughly $\tilde{O}\left(\frac{1}{\varepsilon^2}\right)$ queries provided that $\gamma$ is mildly small (polynomially small in $k$).

Following that, Kontorovich et al. [2022] showed that a qualitatively similar result could be obtained via compression arguments (instead of typical stability). However, their (computationally efficient) algorithms require $\gamma$ to be exponentially small in $k$ and thus do not apply to the parameters $\varepsilon, \gamma$ stated in Question 1.3. They do support other ranges for $(\varepsilon, \gamma)$, but at any case their efficient algorithms cannot answer more than $O(n)$ queries when the parameters $(\varepsilon, \gamma)$ adhere to the behavior of Hoeffding's inequality for i.i.d. samples. For example, for $\varepsilon = O(1)$ and $\gamma = 2^{-\Omega(n)}$ their algorithm supports $O(n)$ adaptive queries to within constant accuracy (even when there are correlations in the data).

To summarize, currently there are two existing techniques for answering adaptively chosen concentrated queries: Either via typical stability in the *small $\varepsilon$ regime* or via compression arguments in the *tiny $\gamma$ regime*. At any case, all known results do not break the $O(n)$ queries barrier, even when the concentration parameters reflect the behavior guaranteed in the i.i.d. setting. In contrast, without correlations in the data, answering $O(n^2)$ queries is possible.

## 1.1 Our results

### 1.1.1 An impossibility result for answering concentrated queries

We establish a new negative result providing strong evidence that the *linear barrier* discussed above is inherent. Our result applies to mechanisms that perturb the empirical mean of a query either by adding independent noise or by evaluating it on a randomly selected subsample of the dataset, which together constitute all known efficient, polynomial-time techniques for answering a super-linear number of queries in the i.i.d. setting. For brevity, we refer to these as **Noise and Subsampling (NS)** mechanisms. Specifically, we show that NS mechanisms cannot answer more than $O(n)$ adaptively chosen concentrated queries, even if the query concentration matches the behavior of Hoeffding's inequality for i.i.d. samples. This constitutes the first negative result for answering adaptively chosen concentrated queries, and stands in sharp contrast to the $O(n^2)$ achievable in the i.i.d. setting. Specifically,

**Theorem 1.4** (informal). *Let $\varepsilon > 0$ and $\gamma \in (0,1]$. Then there exists a domain $\mathcal{X}$ and a distribution $\mathcal{D}$ over $\mathcal{X}^n$ such that the following holds. For any NS mechanism $\mathcal{M}$ there exists an adaptive analyst*

*issuing $(\varepsilon, \gamma)$-concentrated queries $q_1, \ldots, q_k$, where $k = \Omega\left(\min\left\{\frac{1}{\gamma}, \frac{1}{\varepsilon^2}\ln\left(\frac{1}{\varepsilon\cdot\gamma}\right)\right\}\right)$, such that with probability at least $0.9$ there is a query $q_i$ for which the answer provided by $\mathcal{M}$ deviates from its true mean $q_i(\mathcal{D})$ by at least 0.9.*

To interpret this result, note for example that when $\varepsilon = O(1)$ and $\gamma = 2^{-n}$ then our bound on $k$ gives $k = O(n)$. We show that the same is true for all values of $(\varepsilon, \gamma)$ that match the behavior of the Hoeffding bound in the i.i.d. setting.

This negative result emphasizes a fundamental limitation. In order to break the linear barrier on the number of supported queries, future work must either impose additional structural assumptions on the problem or introduce new algorithmic techniques beyond noise addition and subsampling mechanisms.

### 1.1.2 A simplified positive result

As we mentioned, Bassily and Freund [2016] introduced the notion of *typical stability* and leveraged it to design algorithms supporting adaptive concentrated queries. However, their definitions and techniques are quite complex. In particular, bounding the number of supported queries $k$ as a function of the concentration parameters $\varepsilon$ and $\gamma$ is not easily extractable from their theorems.

We present a significantly simpler analysis for their algorithm that does not use typical stability at all. Instead, it relies on techniques from differential privacy Dwork et al. [2006]. In addition to being simpler, our analysis allows us to save logarithmic factors in the resulting bounds on $k$. Formally, we show the following theorem.

**Theorem 1.5** (informal). *Fix parameters $\varepsilon, \gamma$. There exists a noise addition mechanism $\mathcal{M}$ that guarantees $\left(\frac{1}{100}, \frac{1}{100}\right)$-statistical accuracy against any analyst $\mathcal{A}$ issuing at most $k$ queries which are $(\varepsilon, \gamma)$-concentrated, provided that $k = O\left(\min\left\{\frac{1}{\gamma}, \frac{1}{\varepsilon^2\,[\ln(1/\varepsilon)]^2}\right\}\right).$*

In retrospect, leveraging differential privacy (DP) to answer concentrated queries (as we do in this work) is a natural approach as it is simpler than prior work on this topic and aligns with other works on other variants of the ADA problem. In a sense, the reason for the additional complexity in the work of Bassily and Freund [2016] steams from their alternative stability notion (typical stability). To the best of our knowledge, our work is the first to derive meaningful positive results for answering adaptively chosen concentrated queries via differential privacy *when correlations are present in the data*.

### 1.1.3 Technical overview of our negative result (informal)

The key insight underlying our negative result is that query concentration alone does not prevent an attacker from extracting substantial information about correlated data. We consider a domain $\mathcal{X}$ partitioned into $\frac{1}{\varepsilon}$ subsets, and define a distribution $\mathcal{D}$ over $\mathcal{X}^n$ in which each sample consists of $\frac{1}{\varepsilon}$ distinct elements, each drawn from a different subset and repeated $\varepsilon n$ times.

This structure simultaneously maximizes the information each query can reveal while ensuring that every query remains tightly concentrated. The attacker designs each query to assign nonzero values only within a single targeted subset, keeping the empirical mean within $[0, \varepsilon]$ and satisfying $(\varepsilon, \gamma)$-concentration by construction. Yet, the responses still leak significant information about the data.

Building on the adaptive attack of Nissim et al. [2018] for the i.i.d. setting, our attacker progressively identifies the repeated elements: each query randomly assigns binary values within the targeted subset and updates an accumulated score to isolate the correct element. We present a simple analysis adapted to our setting, showing that the sample can be recovered with high probability. This breaks the accuracy guarantee of *any* NS mechanism once the number of queries exceeds our derived lower bound.

Our construction highlights that when correlations are present, concentration alone cannot prevent information leakage. Thus, accurately answering more than a linear number of adaptive, concentrated queries requires stronger structural assumptions on the distribution.

### 1.1.4 Technical overview of our positive result (informal)

We prove Theorem 1.5 by showing that the mechanism that answers queries with their noisy empirical average guarantees statistical accuracy (for an appropriately calibrated noise distribution). To show this, we introduce a thought experiment involving three mechanisms, all initialized with the same sample $S \sim \mathcal{D}$, all interacting with the same analyst $\mathcal{A}$:

- **Real-world mechanism:** Answers each query using the *empirical mean* plus independent noise. This is the mechanism whose accuracy we want to analyze.

- **Oracle mechanism:** Answers each query using its *true mean* under the target distribution $\mathcal{D}$, plus independent noise. Note that this mechanism "knows" the target distribution $\mathcal{D}$. This is not a real mechanism; it only exists as part of our proof. The noise magnitude will be small enough such that this mechanism remain accurate.

- **Hybrid mechanism:** Initially behaves like the real-world mechanism, but switches permanently to behave like the oracle mechanism if at some point the empirical mean on any query deviates significantly from its true mean. This is also not a real mechanism; it exists only as part of our proof.

Our analysis proceeds in two steps. First, we leverage techniques from differential privacy to demonstrate that the output distributions of the the oracle and hybrid mechanisms are close. This allows us to invoke advanced-composition-like theorems from differential privacy, ensuring that the outcome distributions of these two mechanisms remain close even after $k$ adaptive queries.

Second, we identify a class of *good interactions*. In these scenarios, the hybrid mechanism never switches to oracle responses, making its behavior *identical* to the real-world mechanism. We show that these good interactions occur with high probability under the oracle mechanism, and by extension, under the hybrid mechanism. We thus get that the real-world mechanism is also likely to produce these good interactions.

By combining these insights, we conclude that, under suitable concentration assumptions on the queries, the real-world mechanism's outputs closely track those of the oracle mechanism, which is statistically accurate by definition, thus ensuring statistical accuracy even in adaptive settings.

## 2 Preliminaries

We now formally define the two classes of mechanisms for which our negative result holds, which we refer to collectively as **Noise and Subsampling (NS)** mechanisms.

**Definition 2.1** (Noise-Addition Mechanism). *A mechanism $\mathcal{M}$ is a* noise-addition mechanism *if, given a dataset $S$ and a statistical query $q$, it returns $a = q(S) + \eta$, where $\eta$ is a random variable drawn independently from a fixed, zero-mean noise distribution that does not depend on $S$ or $q$.*

**Definition 2.2** (Subsampling Mechanism). *A mechanism $\mathcal{M}$ is a* subsampling mechanism *if, given a dataset $S$ of size $n$, it answers each query $q$ as follows. For each round independently, the mechanism samples a subsample $S'$ of size $m$ by drawing $m$ elements from $S$ independently and uniformly at random with replacement, and returns $a = q(S') = \frac{1}{m} \sum_{x \in S'} q(x)$, the empirical mean of $q$ on the subsample. The subsample $S'$ is freshly resampled for every query, independently of previous rounds.*

**Differential privacy.** Consider an algorithm that operates on a dataset. Differential privacy is a stability notion requiring the (outcome distribution of the) algorithm to be insensitive to changing one example in the dataset. Formally,

**Definition 2.3** (Dwork et al. [2006]). *Let $\mathcal{M}$ be a randomized algorithm whose input is a dataset. Algorithm $\mathcal{M}$ is $(\varepsilon, \delta)$-differentially private (DP) if for any two datasets $S, S'$ that differ in one point (such datasets are called* neighboring*) and for any event $E$ it holds that $\Pr[\mathcal{M}(S) \in E] \leq e^{\varepsilon} \cdot \Pr[\mathcal{M}(S') \in E] + \delta$.*

The most basic constructions of differentially private algorithms are via the Laplace mechanism as follows.

**Definition 2.4** (The Laplace Distribution). *A random variable has probability distribution $\mathrm{Lap}(b)$ if its probability density function is $f(x) = \frac{1}{2b} \exp\left(-\frac{|x|}{b}\right)$, where $x \in \mathbb{R}$.*

**Theorem 2.5** ([Dwork et al.](), [2006]). *Let $f$ be a function that maps datasets to the reals with sensitivity $\ell$ (i.e., for any neighboring $S, S'$ we have $|f(S) - f(S')| \leq \ell$). The mechanism $\mathcal{M}$ that on input $S$ adds noise with distribution $\mathrm{Lap}(\frac{\ell}{\varepsilon})$ to the output of $f(S)$ preserves $(\varepsilon, 0)$-differential privacy.*

**Finite precision and bounded outputs** Real-world computing devices can only produce finitely many bits of precision. Accordingly, we assume outputs are rounded or truncated, ensuring a discrete output space. In our negative result, we additionally assume that outputs lie within a fixed bounded range. This holds automatically for subsampling mechanisms, since the empirical mean of values in $[0, 1]$ always lies in $[0, 1]$. For noise-addition mechanisms, we assume outputs lie in the interval $[-1, 2]$. Since queries are bounded in $[0, 1]$ and the accuracy parameter $\alpha$ is also in $[0, 1]$, any response outside this range would already violate the accuracy guarantee, meaning the mechanism has failed and the attack has succeeded.

## 3 An impossibility result for answering concentrated queries

We begin by noting that the bound of $1/\gamma$ queries is unavoidable. To see this, consider a distribution $\mathcal{D}$ that is uniform over $1/\gamma$ *disjoint* samples $S_1, \ldots, S_{1/\gamma}$ of size $n$ each. Now consider the analyst that queries (one by one) all $1/\gamma$ queries of the form $q_i(x) = 1$ if $x \in S_i$ and $q_i(x) = 0$ otherwise. The "true mean" of each of these queries over $\mathcal{D}$ is exactly $\gamma$, and each of them, the probability of deviating from this true mean by more than $\gamma$ (over sampling $S \sim \mathcal{D}$) is at most $\gamma$. So for $\gamma < \varepsilon$ and $\alpha < 1 - \gamma$ these queries are all concentrated, and one of them causes the mechanism to lose accuracy. See Appendix A.1 for the formal details.

The main result of this section gives a stronger impossibility bound. We construct a distribution and domain such that any NS mechanism can be forced to fail with probability $1 - \beta$ by an attacker issuing only $(\varepsilon, \gamma)$-concentrated queries after $k = \Omega\left(\frac{1}{\varepsilon^2} \cdot \ln\left(\frac{1}{\varepsilon \cdot \beta \cdot \gamma}\right)\right)$ rounds.

We then consider the setting where $\gamma$ is a function of $n$ and $\varepsilon$ that corresponds to the concentration of bounded queries in an i.i.d. regime. Specifically, if $\gamma(n, \varepsilon) = 2\exp(-2\varepsilon^2 n)$, as given by the double-sided Hoeffding inequality, then the two combined bounds imply that no NS mechanism can answer more than $O(n)$ such queries.

**Domain and distribution.** We now formalize the domain and sample construction described above. Let $\varepsilon \in (0, 1]$ and define $r = 1/\varepsilon$. Let $\mathcal{X}$ be a finite domain of size $N = \max\left\{\frac{1}{\varepsilon^2}, \frac{1}{\varepsilon \gamma}\right\}$. Assume for simplicity that $r$ is an integer and that it divides both $N$ and $n$. Partition $\mathcal{X}$ into $r$ disjoint subsets $\mathcal{X}_1, \ldots, \mathcal{X}_r$, each of size $\varepsilon N$. Label the elements in each subset arbitrarily: $\mathcal{X}_i = \{x_i^1, x_i^2, \ldots, x_i^{\varepsilon N}\}$ for all $i = 1, \ldots, r$.

The distribution $\mathcal{D}$ over $\mathcal{X}^n$ is defined as follows. First, sample an index $j \sim \mathrm{Unif}\{1, 2, \ldots, \varepsilon N\}$. Then, output the sample $S = (\underbrace{x_1^j, \ldots, x_1^j}_{\varepsilon n}, \underbrace{x_2^j, \ldots, x_2^j}_{\varepsilon n}, \ldots, \underbrace{x_r^j, \ldots, x_r^j}_{\varepsilon n})$. That is, a single index $j$

determines one point $x_i^j$ from each subset $\mathcal{X}_i$, and the sample consists of each of these points repeated exactly $n/r = \varepsilon n$ times. Although $\mathcal{D}$ is defined on $\mathcal{X}^n$, its support contains only $N/r = \varepsilon N$ distinct samples, and each is determined by the shared index $j$.

**Attack Overview:** The attack procedure (Algorithm 1) operates over $k$ rounds of information gathering followed by a single final query. During the information gathering rounds, each query $q_t$ is constructed using i.i.d. Bernoulli random variables: each element $x_1^j$ in the targeted subset $\mathcal{X}_1$ independently takes the value 1 with probability $p_t \sim \mathrm{Unif}[0, 1]$, while all other elements in the domain receive value 0. Throughout the interaction, we track an accumulated score $Z_j$ for each element $x_1^j$, defined so that the score increment $z_t^j$ has positive expectation if $x_1^j$ matches the unique element appearing in the true sample, and zero expectation otherwise. After all $k$ rounds, we identify the element with the highest cumulative score. By standard concentration arguments, this element is likely to be the one present in the actual sample, as it uniquely accumulates a positive expected score. We issue a final query that evaluates to 1 for the elements in the sample we identified and 0 for all other elements, thus pinpointing the true sample. Throughout the analysis, we assume

$\alpha < 1 - \frac{1}{|\text{Supp}(\mathcal{D})|}$, ensuring that if the final query successfully identifies the true sample, the resulting deviation exceeds $\alpha$.

---

**Algorithm 1** Attack Procedure

---

**Initialization:** Let $\mathcal{M}$ be an NS mechanism initialized with a sample $S \sim \mathcal{D}$. For each element $x_1^j \in \mathcal{X}_1$, initialize an accumulated score $Z_j = 0$.

**Information gathering rounds:** For each round $t \in [k]$:

1. Sample $p_t \sim \text{Unif}[0, 1]$.

2. Define the query $q_t : \mathcal{X} \to \{0, 1\}$ as:

$$q_t(x) = \begin{cases} \sim \text{Bernoulli}(p_t) & \text{if } x \in \mathcal{X}_1, \\ 0 & \text{otherwise.} \end{cases}$$

3. Submit $q_t$ to the mechanism and receive the response $a_t$.

4. For each $x_1^j \in \mathcal{X}_1$, define $z_t^j = (a_t - p_t/r)\,(q_t(x_1^j) - p_t)$, and update $Z_j \leftarrow Z_j + z_t^j$.

**Final query:** After $k$ rounds, compute $j^* = \arg\max_j Z_j$. Submit a final query $q^* : \mathcal{X} \to \{0, 1\}$ by setting

$$q^*(x) = \begin{cases} 1 & \text{if } x = x_i^{j^*} \text{ for } i \in [r], \\ 0 & \text{otherwise.} \end{cases}$$

---

**Analysis of the attack.** We now prove that the attack described above succeeds using only $(\varepsilon, \gamma)$-concentrated queries. This analysis establishes three components: (1) all $k$ information-gathering queries are $(\varepsilon, \gamma)$-concentrated, (2) the final attack query $q^*$ is also $(\varepsilon, \gamma)$-concentrated, and (3) with high probability, the attack correctly identifies the underlying sample. The proofs for the concentration of the queries are deferred to Appendix A.2.

**Theorem 3.1.** *For the attack to identify the true sample with probability at least $1 - \beta$, it suffices that $k = \Omega\left(\frac{1}{\varepsilon^2}\left[\max\left\{\ln\left(\frac{1}{\varepsilon}\right),\, \ln\left(\frac{1}{\gamma}\right)\right\} + \ln\left(\frac{1}{\beta}\right)\right]\right)$.*

*Proof Sketch.* Consider the variables $z_t^j = (a_t - \frac{p_t}{r})(q_t(x_1^j) - p_t)$. It can be shown (Appendix A.3.1) that these satisfy $\mathbb{E}[z_t^j] = \frac{1}{6r}$ if $j$ corresponds to the true sample, and $0$ otherwise. Summing these variables, define $Z_j = \sum_{t=1}^k z_t^j$. Each $Z_j$ thus accumulates a positive expectation only for the true index $j_s$, and zero otherwise. Using standard concentration inequalities, one obtains that the index maximizing $Z_j$ coincides with the true sample with probability at least $1 - \beta$, provided $k$ satisfies the stated bound. The detailed argument is deferred to Appendix A.3. $\qquad\square$

Combining Theorem 3.1 and the attack in Appendix A.1, we get:

**Theorem 3.2.** *Let $\varepsilon > 0$, $\gamma \in (0, 1]$, and $\beta \in (0, 1)$. Then there exists a domain $\mathcal{X}$ and a distribution $\mathcal{D}$ over $\mathcal{X}^n$ such that for any NS mechanism $\mathcal{M}$, and any $\alpha \in \left(0, 1 - \frac{1}{|\text{Supp}(\mathcal{D})|}\right)$, there exists an analyst issuing $k$ adaptive $(\varepsilon, \gamma)$-concentrated queries with $k = \Omega\left(\min\left\{\frac{1}{\gamma},\, \frac{1}{\varepsilon^2}\ln\left(\frac{1}{\varepsilon \cdot \beta \cdot \gamma}\right)\right\}\right)$, such that $\Pr\left[\exists\, i \in [k] \text{ such that } |M(q_i) - q_i(\mathcal{D})| > \alpha\right] \geq 1 - \beta$.*

**Comparison to the i.i.d. setting** We now compare our query bound to the classical i.i.d. setting, where differentially private mechanisms can answer up to $O(n^2)$ adaptive statistical queries with bounded error. In sharp contrast, our results imply a strong negative statement: even if the query concentration matches the behavior of Hoeffding's inequality for i.i.d. samples, the number of accurately answerable queries by NS mechanisms is tightly bounded by $O(n)$. To make this comparison precise, we assume a fixed failure probability (e.g. $\beta = 0.01$), and let the concentration rate $\gamma(n, \varepsilon)$ follow the double-sided Hoeffding bound: $\gamma(n, \varepsilon) = 2\exp(-2n\varepsilon^2)$. Under this assumption, the bound from theorem 3.2 simplifies to $k = O(n)$ for any $\varepsilon \in (0, 1]$. The full derivation is a straightforward asymptotic calculation, deferred to Appendix A.4.

# 4 A simplified positive result

## 4.1 Setup and definitions

We introduce a thought experiment involving three mechanisms, all initialized with the same sample $S \sim \mathcal{D}$, all interacting with the same adaptive analyst $\mathcal{A}$ over $k$ rounds.

A sample $S \sim \mathcal{D}$ is drawn once and remains hidden from the analyst. The analyst $\mathcal{A}$ issues a sequence of $k$ statistical queries $q_1, \ldots, q_k : \mathcal{X} \to [0,1]$, where each query may depend adaptively on previous queries and responses; $\mathcal{A}$ is assumed to be deterministic without loss of generality, as randomized analysts can be treated by taking expectation over a distribution of deterministic strategies.

The mechanism's responses are based on one of three strategies: the real-world mechanism adds Laplace noise to the empirical mean $\mathcal{M}_S(q_i) = q_i(S) + \eta_i$, where $\eta_i \sim \mathrm{Laplace}(0, b)$; the oracle mechanism adds Laplace noise to the true mean $\mathcal{M}_O(q_i) = q_i(\mathcal{D}) + \eta_i'$, where $\eta_i' \sim \mathrm{Laplace}(0, b)$; and the hybrid mechanism responds as the real-world mechanism while all past queries are $\varepsilon$-concentrated relative to $S$, but switches to the oracle mechanism once any empirical mean deviates by more than $\varepsilon$ from the true mean: $\mathcal{M}_H(q_i) = \begin{cases} q_i(S) + \eta_i, & \text{if } \max_{j \leq i} |\hat{q}_j(S) - q_j(\mathcal{D})| \leq \varepsilon, \\ q_i(\mathcal{D}) + \eta_i', & \text{otherwise.} \end{cases}$

To describe the interaction between the analyst and the mechanism, we define the *transcript* $t = (q_1, a_1, \ldots, q_k, a_k)$, which records the sequence of queries and their corresponding responses. Each answer $a_i$ is given by either $\mathcal{M}_S(q_i)$, $\mathcal{M}_O(q_i)$, or $\mathcal{M}_H(q_i)$, depending on the mechanism being used in the interaction.

**Definition 4.1** (Transcript). *Let $\mathcal{A}$ be an analyst interacting with a mechanism over $k$ rounds. The random transcript $T$ is the sequence of queries and responses generated in the interaction. A particular outcome is denoted by $t = (q_1, a_1, \ldots, q_k, a_k)$.*

**Transcript Probability Notation.** Let $t$ be a transcript arising from an interaction between an analyst $\mathcal{A}$ and a mechanism $\mathcal{M}$. We denote the probability of $t$ arising under mechanism $\mathcal{M}$ as $\Pr_M(T = t)$, where $\Pr_{\mathcal{M}_S}(T = t)$, $\Pr_{\mathcal{M}_O}(T = t)$, and $\Pr_{\mathcal{M}_H}(T = t)$ refer to the probabilities under the real-world, oracle, and hybrid mechanisms, respectively.

To analyze the outcome of the interaction, we define two categories of "good" events: (1) *Statistical accuracy*: This event contains all transcripts $t$ such that all answers in $t$ are close to the true means of their respective queries. (2) *Sample concentration*: This event contains all pairs of transcripts $t$ and samples $S$ such that the empirical mean on $S$ of each query in $t$ is close to its true mean on $\mathcal{D}$. Note that *statistical accuracy* is a property of the mechanism's outputs and their deviation from the true means, independent of the sample; *sample concentration*, by contrast, depends on both transcript and sample as it reflects how well the empirical means align with the true expectations.

**Definition 4.2** ($\alpha$-accurate transcript). *A transcript $t = (q_1, a_1, \ldots, q_k, a_k)$ is $\alpha$-accurate if every response $a_i$ is within $\alpha$ of the true mean $q_i(\mathcal{D})$; that is: $|a_i - q_i(\mathcal{D})| \leq \alpha$ for all $i \in [k]$.*

**Definition 4.3** ($\varepsilon$-good pair $(S, t)$). *Let $S \in \mathrm{Supp}(\mathcal{D})$ be a sample and let $t = (q_1, a_1, \ldots, q_k, a_k)$ be a transcript of $k$ queries and responses. The pair $(S, t)$ is called $\varepsilon$-good if, for every query $q_i$ in $t$, the empirical mean of $q_i$ on $S$ is close to its true mean: $|q_i(S) - q_i(\mathcal{D})| \leq \varepsilon$.*

Our strategy involves demonstrating that the probability of sample concentration events occurring is similar across the real-world, oracle, and hybrid mechanisms. By establishing this, we can infer that events satisfying both statistical accuracy and sample concentration—which occur with high probability under the oracle mechanism—also occur with high probability under the real-world mechanism. Thus ensuring that the real-world mechanism maintains statistical accuracy despite the adaptivity of the analyst.

## 4.2 Relating the distribution of events under the oracle and hybrid mechanisms

The first component of our analysis shows that the output distributions of the oracle and hybrid mechanisms are closely aligned, similarly to the guarantees provided by $(\varepsilon, 0)$-differential privacy for neighboring datasets. This is formalized in Lemma 4.4, with the proof deferred to Appendix B.1

**Lemma 4.4.** *Let $S \in \mathrm{Supp}(\mathcal{D})$ be a sample, and let $q$ be any query. Then for every measurable set in the output space $\mathcal{E} \subseteq \mathcal{Y}$: $\Pr\big[\mathcal{M}_H(q) \in \mathcal{E}\big] \leq e^{\frac{\varepsilon}{b}} \Pr\big[\mathcal{M}_O(q) \in \mathcal{E}\big]$ and $\Pr\big[\mathcal{M}_O(q) \in \mathcal{E}\big] \leq e^{\frac{\varepsilon}{b}} \Pr\big[\mathcal{M}_H(q) \in \mathcal{E}\big]$.*

**Extending advanced composition to our setting.** The preceding lemma allows us to extend the advanced composition analysis of Dwork et al. [2010b] (see also Dwork and Roth [2014]) to our framework. One of their results shows that if an $(\varepsilon, 0)$-differentially private mechanism interacts with an analyst over $k$ rounds, then for any $\delta' > 0$, the overall interaction is $(\varepsilon^*, \delta')$-differentially private, where $\varepsilon^* \approx \sqrt{k}\varepsilon$. Although this theorem is framed in terms of differential privacy and neighboring datasets, the proof relies solely on the following: in each round, the conditional distributions of the outputs in two parallel experiments $Y$ and $Z$ given identical histories up to the previous round satisfy that for any $E \subseteq \mathrm{Supp}(Y)$, it holds that $\ln\left(\frac{\Pr[Y \in E]}{\Pr[Z \in E]}\right) \leq \varepsilon$, and similarly for the reverse ratio. In our setting, Lemma 4.4 implies that this condition holds for any interaction of a fixed analyst with the oracle and hybrid mechanisms once you condition on identical histories. We now formalize the corresponding composition theorem in our framework and, for completeness, supply a full proof in appendix B.2 that mimics the proof of [Dwork et al., 2010b, Theorem III.1] to demonstrate that it applies under our conditions.

**Theorem 4.5.** *Let $S \in \mathrm{Supp}(\mathcal{D})$ be a sample, and let $\mathcal{A}$ be a fixed analyst. Consider two $k$-round interactions with $\mathcal{A}$: one with the hybrid mechanism $\mathcal{M}_H$, and one with the oracle mechanism $\mathcal{M}_O$. For any $\rho > 0$, define $\varepsilon^* = \sqrt{2k\ln\left(\frac{1}{\rho}\right)}\frac{\varepsilon}{b} + k\frac{\varepsilon}{b}\left(e^{\varepsilon/b} - 1\right)$. Then*

$$\Pr_{t \leftarrow \mathcal{M}_H}\left[\ln\frac{\Pr_{\mathcal{M}_H}(T = t)}{\Pr_{\mathcal{M}_O}(T = t)} > \varepsilon^*\right] \leq \rho, \quad \text{and} \quad \Pr_{t \leftarrow \mathcal{M}_O}\left[\ln\frac{\Pr_{\mathcal{M}_O}(T = t)}{\Pr_{\mathcal{M}_H}(T = t)} > \varepsilon^*\right] \leq \rho$$

From this, we conclude the following corollary (the proof is deferred to Appendix B.3):

**Corollary 4.6.** *Let $\mathcal{A}$ be a fixed analyst, $S \in \mathrm{Supp}(\mathcal{D})$ a sample, and let $\mathcal{E}$ be any event that can arise in the interaction with the analyst. For any $\rho > 0$, define $\varepsilon^*$ as in Theorem 4.5. Then $e^{-\varepsilon^*}\left(\Pr_{\mathcal{M}_O}[\mathcal{E}] - \rho\right) \leq \Pr_{\mathcal{M}_H}[\mathcal{E}] \leq e^{\varepsilon^*}\Pr_{\mathcal{M}_O}[\mathcal{E}] + \rho$.*

### 4.3 High-probability accuracy of the Laplace mechanism

The next lemma shows that the real-world and hybrid mechanisms assign equal probability to any event consisting entirely of $\varepsilon$-good sample–transcript pairs. This follows from the fact that both mechanisms operate with the same randomness, so as long as all queries remain $\varepsilon$-good, their responses are identical. Full details are provided in Appendix B.4.

**Lemma 4.7.** *Fix an analyst $\mathcal{A}$, and let $\mathcal{G}$ be the set of $\varepsilon$-good sample–transcript pairs. Then, for every measurable subset $\mathcal{E} \subseteq \mathcal{G}$, $\Pr_{\mathcal{M}_S}[\mathcal{E}] = \Pr_{\mathcal{M}_H}[\mathcal{E}]$.*

The following lemma provides a high-probability guarantee for $\varepsilon$-good and $\alpha$-accurate transcripts under the oracle mechanism. Since the output is independent of the sample, a union bound over queries and noise yields this result, with the full argument deferred to Appendix B.5.

**Lemma 4.8.** *Let $S \sim \mathcal{D}$ and consider a $k$-round interaction between an analyst $\mathcal{A}$ and the oracle mechanism, producing the transcript $t$. Define the failure probability of any of the Laplace noises exceeding $\alpha$ as $\zeta = 1 - \Pr[|\eta_1'| \leq \alpha]^k$, for $\eta_1' \sim \mathrm{Laplace}(0, b)$. Then,*

$$\Pr_{S \sim \mathcal{D}, \, t \sim Pr_{\mathcal{M}_O}}\big[(S, t) \text{ is } \varepsilon\text{-good and } t \text{ is } \alpha\text{-accurate}\big] \geq 1 - k\gamma - \zeta.$$

We have established that: (1) the transcript distribution under the hybrid mechanism closely approximates that of the oracle mechanism, (2) the probability of any event consisting of $\varepsilon$-good sample–transcript pairs is identical under both the real-world and hybrid mechanisms, and (3) under the oracle mechanism, the joint event of $\varepsilon$-good pairs and $\alpha$-accuracy occurs with high probability. Combining these facts implies that the real-world mechanism is statistically accurate with high probability. This is formalized in the following theorem, the proof of which is deferred to Appendix B.6.

**Theorem 4.9.** *Let $\mathcal{A}$ be an analyst, and $\mathcal{M}_S$ the real-world Laplace mechanism interacting with $\mathcal{A}$ over $k$ rounds. For $\alpha > 0$ and $\rho > 0$, define $\varepsilon^*$ as in theorem 4.5, and let $\zeta$ be as in lemma 4.8. Then, the probability that the real-world mechanism produces an $\alpha$-accurate transcript satisfies*

$$\Pr_{\mathcal{M}_S}\left[t = (q_1, a_1, \ldots, q_k, a_k) \,:\, \forall i \, |a_i - q_i(\mathcal{D})| \le \alpha\right] \ge e^{-\varepsilon^*}\left(1 - k\,\gamma - \zeta - \rho\right).$$

**Theorem 4.10.** *Let $\mathcal{A}$ be any analyst issuing $k$ adaptive $(\varepsilon, \gamma)$-concentrated queries, and fix an accuracy parameter $\alpha > 0$ and failure probability $\beta > 0$. Then the Laplace mechanism can achieve $(\alpha, \beta)$-accuracy over all $k$ queries provided $k = O\Big(\min\Big\{\frac{\beta}{\gamma},\ \beta\,\varepsilon^{-2},\ \frac{\alpha^2\,\beta^2}{\varepsilon^2\,[\ln(1/\varepsilon)]^2\,\ln(1/\beta)}\Big\}\Big).$*

*Proof.* Run the Laplace mechanism with noise scale $b = \frac{\alpha}{2\,\ln(1/\varepsilon)}$. Theorem 4.9 implies that for any fixed $\alpha > 0$ and number of queries $k$, the real-world mechanism satisfies $(\alpha, \beta)$-accuracy provided $e^{-\varepsilon^*} \cdot \left(1 - k\,\gamma - \zeta - \rho\right) \ge 1 - \beta$. Requiring each term in the failure probability $\rho$, $\zeta$, $k\,\gamma$ and $\varepsilon^*$ to be $\le \beta/4$, yields the desired result. See Appendix B.7 for the details of this derivation. $\qquad\square$

**Simplified bound for constant accuracy and failure (example).** If we assume constant parameters for failure probability and accuracy with $\varepsilon < \alpha$ (e.g., $\alpha = \beta = 0.01$), Theorem 4.10 implies the existence of a noise addition mechanism $\mathcal{M}$ that guarantees $(0.01, 0.01)$-statistical accuracy against any analyst $\mathcal{A}$ issuing up to $k$ adaptive, $(\varepsilon, \gamma)$-concentrated queries, provided

$$k = O\left(\min\left\{\frac{1}{\gamma},\ \frac{1}{\varepsilon^2[\ln(1/\varepsilon)]^2}\right\}\right).$$

## Acknowledgments and Disclosure of Funding

Emma Rapoport is partially supported by the Israel Science Foundation (grant 1419/24), the Blavatnik Family Foundation and the Deutsch Foundation. Edith Cohen is partially supported by Israel Science Foundation, (grant 1156/23). Uri Stemmer is partially supported by the Israel Science Foundation (grant 1419/24) and the Blavatnik Family Foundation.

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

# Appendices

## A  Additional proofs for negative result

### A.1  A simple negative result using $1/\gamma$ queries

We present a simple construction showing that for any values of $0 < \gamma \le \varepsilon \le 1$ and any $\alpha \le 1 - \gamma$, there exists a domain, distribution, and adversary strategy such that after at most $k = 1/\gamma$ queries, all of which are $(\varepsilon, \gamma)$-concentrated, the mechanism is forced to return a response that is not statistically-accurate.

**Claim A.1.** *Fix parameters $0 < \gamma \le \varepsilon \le 1$ and $\alpha \le 1 - \gamma$. There exists a distribution $\mathcal{D}$ over $\mathcal{X}^n$ and a set of $k = \frac{1}{\gamma}$ queries, each $(\varepsilon, \gamma)$-concentrated, such that an attacker submitting these queries to any mechanism will receive a response that differs from its true expectation by more than $\alpha$ on at least one query.*

*Proof.* Let $r = 1/\gamma$, assumed to be an integer for simplicity. Let the domain be $\mathcal{X} = \{1, 2, \ldots, rn\}$, partitioned into $r$ disjoint subsets $S_1, \ldots, S_r \subset \mathcal{X}$, each of size $n$.

Let $\mathcal{D}$ be the uniform distribution over these subsets: that is, $S \sim \mathcal{D}$ means $S = S_i$ with probability $1/r = \gamma$ for any $i \in \{1, \ldots, r\}$.

Define queries $q_{S_1}, \ldots, q_{S_r} : \mathcal{X} \to \{0, 1\}$ by

$$q_{S_i}(x) = \begin{cases} 1 & \text{if } x \in S_i, \\ 0 & \text{otherwise.} \end{cases}$$

Each query has true mean $q_{S_i}(\mathcal{D}) = 1/r = \gamma$. For any sample $S = S_i$, we have $q_{S_i}(S) = 1$, while for all $j \ne i$, $q_{S_j}(S) = 0$.

To verify concentration, note that for any $S \ne S_i$, The empirical mean $q_{S_i}(S) = 0$, so the deviation from the true mean is exactly $\gamma \le \varepsilon$. For the sample $S = S_i$, $q_{S_i}(S) = 1$, and the deviation is $1 - \gamma \ge \alpha$.

Thus, submitting the $k = 1/\gamma$ queries guarantees that one query must yield an error greater than $\alpha$, violating $(\alpha, \beta)$-statistical accuracy. $\square$

### A.2  All queries in the attack described in algorithm 1 are $(\varepsilon, \gamma)$-concentrated

**Lemma A.2.** *Each query $q_t$ in the first $k$ rounds in the attack described in algorithm 1 is $(\varepsilon, \gamma)$-concentrated.*

*Proof.* Fix round $t$ and any $S' \in \mathrm{Supp}(\mathcal{D})$. The sample consists of $\varepsilon n$ copies of $x_1^j$ of some element $x_1^j \in \mathcal{X}_1$, and $(n - \varepsilon n)$ elements from $\mathcal{X} \setminus \mathcal{X}_1$. Since $q_t(x) = 0$ for $x \notin \mathcal{X}_1$, we have $q_t(S') = \varepsilon q_t(x_1^j)$, where $q_t(x_1^j) \sim \mathrm{Bernoulli}(p_t)$. Therefore the empirical mean of any sample in $\mathrm{Supp}(\mathcal{D})$ is in $\{0, \varepsilon\}$, and $q_t(\mathcal{D})$ the true mean is $\varepsilon p_t \in [0, \varepsilon]$, so the absolute deviation is at most $\varepsilon$. $\square$

**Lemma A.3.** *The final query $q^*$ in the attack described in algorithm 1 is $(\varepsilon, \gamma)$-concentrated under $\mathcal{D}$.*

*Proof.* Let $T \sim \mathcal{D}$. The query $q^*$ evaluates to 1 if $T$ is the sample generated by choosing the $j^*$-th element in each subset $\mathcal{X}_i$, and 0 otherwise. Thus, the true mean $q^*(\mathcal{D}) = \frac{1}{\varepsilon N}$. If $q^*(T) = 0$, the deviation is $\frac{1}{\varepsilon N}$; if $q^*(T) = 1$, the deviation exceeds $\varepsilon$ but this event occurs with probability $\frac{1}{\varepsilon N}$. Since $N = \max\left\{\frac{1}{\varepsilon^2}, \frac{1}{\varepsilon \gamma}\right\}$, we have $\frac{1}{\varepsilon N} \le \varepsilon$ and $\frac{1}{\varepsilon N} \le \gamma$, so the deviation exceeds $\varepsilon$ with probability at most $\gamma$, meaning $q^*$ is $(\varepsilon, \gamma)$-concentrated. $\square$

### A.3  Proof of theorem 3.1

We begin by proving a supporting lemma:

### A.3.1 Lemma A.4

**Lemma A.4.** *Let $j_s$ denote the index of the elements that appear in the true sample $S$ that is used by the mechanism. Define*

$$z_t^j = \left(a_t - \frac{p_t}{r}\right)\left(q_t(x_1^j) - p_t\right).$$

*Then for each $j \in \{1, \ldots, \varepsilon N\}$ and $t \in \{1, \ldots, k\}$,*

$$\mathbb{E}[z_t^j] = \begin{cases} \frac{1}{6r} & \text{if } j = j_s, \\ 0 & \text{otherwise.} \end{cases}$$

*Proof.* Let $S = (x_1^{j_s}, \ldots, x_1^{j_s}, \ldots, x_r^{j_s} \ldots, x_r^{j_s})$ be the true sample that is used by the mechanism. *Case 1: $j \neq j_s$.* Here $q_t(x_1^j)$ is independent of $a_t$, and $\mathbb{E}[q_t(x_1^j)] = p_t$, so:

$$\mathbb{E}[z_t^j] = \mathbb{E}[(a_t - \tfrac{p_t}{r})(q_t(x_1^j) - p_t)] = 0.$$

*Case 2: $j = j_s$.* The analysis in this case depends on the type of mechanism used.

If $\mathcal{M}$ is a **noise addition mechanism:** Substitute $a_t = \frac{q_t(x_1^{j_s})}{r} + \eta_t$ into $z_t^{j_s}$. Since $\eta_t$ is zero-mean and independent,

$$\mathbb{E}[z_t^{j_s}] = \frac{1}{r}\mathbb{E}[(q_t(x_1^{j_s}) - p_t)^2].$$

Because $q_t(x_1^{j_s}) \sim \text{Bernoulli}(p_t)$ and $p_t \sim \text{Unif}[0,1]$, this gives:

$$\mathbb{E}[z_t^{j_s}] = \frac{1}{r} \cdot \mathbb{E}[p_t(1 - p_t)] = \frac{1}{6r}.$$

If $\mathcal{M}$ is a **subsampling mechanism:** The output $a_t$ is the empirical mean of the query evaluated on a subsample $S_M$ of size $m$ which is constructed by drawing elements i.i.d. and uniformly from the original sample $S$. We can express the output as:

$$a_t = q_t(S_M) = \frac{1}{m}\sum_{\ell=1}^m q_t(Y_\ell),$$

where each $Y_\ell$ is drawn uniformly from $S$. By construction, the original sample $S$ (of size $n$) contains exactly $n/r$ copies of $x_1^{j_s}$. Therefore, the probability that any given draw $Y_\ell$ is equal to $x_1^{j_s}$ is:

$$\Pr(Y_\ell = x_1^{j_s}) = \frac{n/r}{n} = \frac{1}{r}.$$

Furthermore, $q_t(Y_\ell) = 0$ whenever $Y_\ell \neq x_1^{j_s}$. Let $C$ be the number of times $x_1^{j_s}$ is drawn into $S_M$, so $C \sim \text{Bin}(m, 1/r)$. The output can then be simplified to:

$$a_t = \frac{C}{m}q_t(x_1^{j_s}),$$

as all other points sampled into $S_M$ contribute zero to the sum. We first compute the expectation conditioned on $p_t$ and $C$:

$$\mathbb{E}\left[(a_t - \tfrac{p_t}{r})(q_t(x_1^{j_s}) - p_t) \,\Big|\, p_t, C\right] = \mathbb{E}\left[\left(\frac{C}{m}q_t(x_1^{j_s}) - \frac{p_t}{r}\right)(q_t(x_1^{j_s}) - p_t)\,\Big|\, p_t, C\right]$$

$$= \mathbb{E}\left[\frac{C}{m}q_t(x_1^{j_s})\big(q_t(x_1^{j_s}) - p_t\big)\,\Big|\, p_t\right] - \mathbb{E}\left[\frac{p_t}{r}\big(q_t(x_1^{j_s}) - p_t\big)\,\Big|\, p_t\right]$$

$$= \frac{C}{m}p_t(1 - p_t).$$

Next, taking the expectation over the binomial random variable $C$ yields the expectation conditioned only on $p_t$:

$$\mathbb{E}[z_t^{j_s} \mid p_t] = \mathbb{E}\left[\frac{C}{m}\right]p_t(1 - p_t) = \frac{1}{r}p_t(1 - p_t).$$

Thus, in the subsampling case as well, the final expectation is:

$$\mathbb{E}[z_t^{j_s}] = \frac{1}{r}\mathbb{E}[p_t(1 - p_t)] = \frac{1}{6r}.$$

$\square$

**Theorem 3.1.** *For the attack to identify the true sample with probability at least $1 - \beta$, it suffices that*

$$k = \Omega\left(\frac{1}{\varepsilon^2}\left[\max\left\{\ln\left(\frac{1}{\varepsilon}\right), \ln\left(\frac{1}{\gamma}\right)\right\} + \ln\left(\frac{1}{\beta}\right)\right]\right).$$

*Proof of theorem 3.1.* Let $j_s$ denote the index of the true sample fed to the mechanism. Define for each index $j$ the cumulative variable $Z_j = \sum_{t=1}^{k} z_t^j$, where

$$z_t^j = \left(a_t - \frac{p_t}{r}\right)\left(q_t(x_1^j) - p_t\right).$$

According to lemma A.4:

$$\mathbb{E}[z_t^j] = \begin{cases} \frac{1}{6r} & \text{if } j = j_s, \\ 0 & \text{otherwise.} \end{cases}$$

For each $j \neq j_s$, define the difference $W_t^j = z_t^{j_s} - z_t^j$, and let $W^j = \sum_{t=1}^{k} W_t^j = Z_{j_s} - Z_j$. Note that for a fixed $j$, the variables $z_1^j, \ldots z_k^j$ are i.i.d., and so are the variables $W_1^j, \ldots, W_k^j$.

By assumption, mechanism outputs are bounded in a fixed interval. Since $p_t, q_t(x) \in [0, 1]$, each term $|W_t^j|$ is bounded. Also, $\mathbb{E}[W_t^j] = \frac{1}{6r}$.

Applying Hoeffding's inequality to $W^j$, we get:

$$\Pr[W^j \leq 0] \leq \exp\left(-\frac{k}{Cr^2}\right),$$

for some constant $C$.

We compare each alternative index $j \neq j_s$ to the true one by checking whether $Z_j \geq Z_{j_s}$, which occurs exactly when $W^j \leq 0$. By a union bound over the $N/r - 1$ such indices:

$$\Pr\left[\exists j \neq j_s : Z_j \geq Z_{j_s}\right] \leq \frac{N}{r} \cdot \exp\left(-\frac{k}{Cr^2}\right).$$

To ensure that the attack identifies the true sample with probability of at least $1 - \beta$, it suffices that

$$\frac{N}{r} \cdot \exp\left(-\frac{k}{Cr^2}\right) \leq \beta \quad \Rightarrow \quad k \geq Cr^2 \cdot \ln\left(\frac{N}{r\beta}\right).$$

Substituting $r = 1/\varepsilon$ and using the definition of $N = \max\left\{\frac{1}{\varepsilon^2}, \frac{1}{\varepsilon\gamma}\right\}$, we obtain

$$\ln\left(\frac{N}{r\beta}\right) = \ln(\varepsilon N) + \ln\left(\frac{1}{\beta}\right) = \max\left\{\ln\left(\frac{1}{\varepsilon}\right), \ln\left(\frac{1}{\gamma}\right)\right\} + \ln\left(\frac{1}{\beta}\right).$$

Thus,

$$k = \Omega\left(\frac{1}{\varepsilon^2}\left[\max\left\{\ln\left(\frac{1}{\varepsilon}\right), \ln\left(\frac{1}{\gamma}\right)\right\} + \ln\left(\frac{1}{\beta}\right)\right]\right),$$

$\square$

## A.4 Comparison to the i.i.d. setting

**Fixed failure probability.** For simplicity, we assume the failure probability $\beta$ is a constant (e.g. $\beta = 0.01$). Under this simplification, the final bound from Section 3 means that no noise-addition mechanism can maintain $(\alpha, \beta)$-statistical accuracy for:

$$k = \min\left\{\frac{1}{\gamma}, \; O\left(\frac{1}{\varepsilon^2}\max\left\{\ln\left(\frac{1}{\varepsilon}\right), \ln\left(\frac{1}{\gamma}\right)\right\}\right)\right\}.$$

**Comparison under Hoeffding-style concentration** To mirror the i.i.d. setting, let $\gamma(n, \varepsilon)$ be the double-sided Hoeffding bound: $\gamma(n, \varepsilon) = 2\exp(-2n\varepsilon^2)$. This yields

$$k = \min\left\{\frac{1}{2}\exp(2n\varepsilon^2), \; O\left(\frac{1}{\varepsilon^2}\max\left\{\ln\left(\frac{1}{\varepsilon}\right), n\varepsilon^2\right\}\right)\right\}.$$

**Parametrizing the concentration rate** To understand how $k$ scales with $n$, we write $\varepsilon(n) = f(n)/\sqrt{n}$, where $f(n) \in (0, \sqrt{n}]$. This gives:

$$n\varepsilon^2 = f(n)^2, \quad \ln\left(\frac{1}{\varepsilon}\right) = \frac{1}{2}\ln n - \ln f(n).$$

Substituting, we get:

$$\frac{1}{\gamma(n,\varepsilon)} = \frac{1}{2}\exp(2f(n)^2), \quad \frac{1}{\varepsilon^2}\max\left\{\ln\left(\frac{1}{\varepsilon}\right), n\varepsilon^2\right\} = \frac{n}{f(n)^2}\cdot\max\left\{\ln\left(\frac{\sqrt{n}}{f(n)}\right), f(n)^2\right\}.$$

We divide the analysis into three regimes based on the value of $f(n)^2$ relative to $\ln n$:

**Case 1:** $f(n)^2 > \ln n$. In this case, the maximum in the second term is attained by $f(n)^2$, so:

$$k = O\left(\frac{n}{f(n)^2}\cdot f(n)^2\right) = O(n).$$

**Case 2:** $f(n)^2 \in [\frac{1}{2}\ln n, \ln n]$

$$k = O\left(\frac{n}{\ln n}\cdot \ln n\right) = O(n).$$

**Case 3:** $f(n)^2 < \frac{1}{2}\ln n$

$$\frac{1}{\gamma(n,\varepsilon)} = \frac{1}{2}\exp(2f(n)^2) \leq \frac{1}{2}\exp(\ln n) = O(n).$$

**Conclusion** For any choice of $\varepsilon \in (0,1]$, whether constant or varying with $n$ (e.g., $\varepsilon(n) = \frac{1}{\sqrt{n}}$), if $\gamma(n,\varepsilon)$ matches the behavior of the Hoeffding concentration bound, the resulting bound is $k = O(n)$.

# B  Additional proofs for positive result

## B.1  Proof of lemma 4.4

**Lemma 4.4.** *Let $S \in \mathrm{Supp}(\mathcal{D})$ be a sample, and let $q$ be any query. Then for every measurable set in the output space $\mathcal{E} \subseteq \mathcal{Y}$: $\Pr[\mathcal{M}_H(q) \in \mathcal{E}] \leq e^{\frac{\varepsilon}{b}}\Pr[\mathcal{M}_O(q) \in \mathcal{E}]$ and $\Pr[\mathcal{M}_O(q) \in \mathcal{E}] \leq e^{\frac{\varepsilon}{b}}\Pr[\mathcal{M}_H(q) \in \mathcal{E}]$.*

*Proof of lemma 4.4.* If $\mathcal{M}_H(q) = \mathcal{M}_O(q)$, the probabilities are equal. Otherwise, since $\mathcal{M}_H(q) = \mathcal{M}_S(q)$ and $|q(S) - q(\mathcal{D})| \leq \varepsilon$, we have $\mathcal{M}_H(q) \sim \mathrm{Laplace}(q(S), b)$ and $\mathcal{M}_O(q) \sim \mathrm{Laplace}(q(\mathcal{D}), b)$. The two distributions differ only in location by at most $\varepsilon$. Therefore, their density ratio is bounded: $\frac{p_{\mathcal{M}_H(q)}(y)}{p_{\mathcal{M}_O(q)}(y)} \leq \exp(\varepsilon/b)$ for all $y \in \mathbb{R}$, and similarly for the reverse ratio. Assuming the mechanism outputs are discretized to a finite set $\mathcal{Y} \subset \mathbb{R}$ by rounding to fixed precision, each output value $y \in \mathcal{Y}$ corresponds to an interval $I_y \subset \mathbb{R}$. Integrating over these intervals preserves the density ratio bound, yielding the stated probability bounds. $\square$

## B.2  Proof of Theorem 4.5

Before presenting the full proof of theorem 4.5, we first introduce additional preliminaries, notation, and a supporting lemma that are used throughout the argument.

### B.2.1  Additional preliminaries

**Definition B.1** (KL divergence or relative entropy [Kullback and Leibler, 1951])**.** *For two distributions $Y$ and $Z$ on the same domain, the* KL divergence *(or* relative entropy*) of $Y$ from $Z$ is*

$$D(Y\|Z) = \mathbb{E}_{y\sim Y}\left[\ln\left(\frac{\Pr[Y=y]}{\Pr[Z=y]}\right)\right].$$

We now recall several definitions and results from Dwork et al. [2010b] that are instrumental in the proof of advanced composition in differential privacy, and which we will use directly in our analysis.

**Definition B.2** (Max divergence, e.g., [Dwork et al., 2010a]). *Let $Y$ and $Z$ be distributions on the same domain. Their* max divergence *is*

$$D_\infty(Y\|Z) \;=\; \max_{S \subseteq \mathrm{Supp}(Y)} \ln\!\Big(\tfrac{\Pr[Y \in S]}{\Pr[Z \in S]}\Big).$$

**Lemma B.3** (Lemma III.2 in Dwork et al. [2010b]). *If $Y$ and $Z$ satisfy $D_\infty(Y\|Z) \le \varepsilon$ and $D_\infty(Z\|Y) \le \varepsilon$, then $D(Y\|Z) \le \varepsilon\,(e^\varepsilon - 1)$.*

**Lemma B.4** (Azuma–Hoeffding inequality [Azuma, 1967]). *Let $C_1, \ldots, C_k$ be real-valued random variables with $|C_i| \le a$ almost surely. Suppose also that $\mathbb{E}[C_i \mid C_1 = c_1, \ldots, C_{i-1} = c_{i-1}] \le \beta$ for every partial sequence $(c_1, \ldots, c_{i-1}) \in \mathrm{Supp}(C_1, \ldots, C_{i-1})$. Then, for any $z > 0$,*

$$\Pr\Big[\sum_{i=1}^{k} C_i > k\,\beta + z\,\sqrt{k}\,a\Big] \;\le\; e^{-z^2/2}.$$

### B.2.2 Definitions and notations

We recall the definition of a transcript:

**Definition 4.1** (Transcript). *Let $\mathcal{A}$ be an analyst interacting with a mechanism over $k$ rounds. The random transcript $T$ is the sequence of queries and responses generated in the interaction. A particular realization is denoted by $t = (q_1, a_1, \ldots, q_k, a_k)$.*

**Extended notation.** We extend the transcript notation introduced above by letting $Q_i$ and $A_i$ denote the random variables corresponding to the query issued and response returned at round $i$, respectively. The full transcript is then the random tuple $T = (Q_1, A_1, \ldots, Q_k, A_k)$, and a specific realization is written $t = (q_1, a_1, \ldots, q_k, a_k)$. The values of $A_i$ depend on the mechanism: in the real-world mechanism, $A_i = \mathcal{M}_S(q_i)$; in the oracle mechanism, $A_i = \mathcal{M}_O(q_i)$; and in the hybrid mechanism, $A_i = \mathcal{M}_H(q_i)$.

**Definition B.5** (Transcript prefix). *For each round $i \in [k]$, the prefix of the transcript up to round $i$ is the random variable $T_{i-1} = (Q_1, A_1, \ldots, Q_{i-1}, A_{i-1})$. For a particular realization $t = (q_1, a_1, \ldots, q_k, a_k)$, we write the corresponding prefix as $t_{i-1} = (q_1, a_1, \ldots, q_{i-1}, a_{i-1})$.*

**Definition B.6** (Support of transcripts). *Let $\mathcal{D}$ be the data distribution over $\mathcal{X}^n$, and let $T$ be the random transcript produced by an interaction (with any of the mechanisms) with an analyst $\mathcal{A}$. We define: $\mathcal{T}_\mathcal{A} = \{\, t : \Pr[T = t] > 0 \,\}$.*

**Remark B.7.** *The support $\mathcal{T}_\mathcal{A}$ depends only on the analyst $\mathcal{A}$, not on the mechanism. This is true because all mechanisms respond by adding independent Laplace noise to either $q_i(S)$ or $q_i(\mathcal{D})$, and, by assumption, the output space $\mathcal{Y}$ is finite. Therefore, for any fixed query $q_i$, every output $a_i \in \mathcal{Y}$ occurs with positive probability under all mechanisms. As a result, the transcript $t = (q_1, a_1, \ldots, q_k, a_k)$ has nonzero probability under each mechanism if and only if it is possible under the analyst's query selection behavior.*

### B.2.3 Supporting Lemma

**Lemma B.8.** *Let $S \in \mathrm{Supp}(\mathcal{D})$ be a fixed sample, and let $q$ be any query. Then the Kullback–Leibler divergence between the outputs of the hybrid and oracle mechanisms satisfies*

$$D\big(\mathcal{M}_H(q) \,\|\, \mathcal{M}_O(q)\big) \le \frac{\varepsilon}{b} \cdot \big(e^{\varepsilon/b} - 1\big), \qquad D\big(\mathcal{M}_O(q) \,\|\, \mathcal{M}_H(q)\big) \le \frac{\varepsilon}{b} \cdot \big(e^{\varepsilon/b} - 1\big).$$

*Proof.* By Lemma 4.4, for every measurable event $E \subseteq \mathcal{Y}$, we have

$$\frac{\Pr[\mathcal{M}_H(q) \in E]}{\Pr[\mathcal{M}_O(q) \in E]} \le \exp(\varepsilon/b), \qquad \frac{\Pr[\mathcal{M}_O(q) \in E]}{\Pr[\mathcal{M}_H(q) \in E]} \le \exp(\varepsilon/b).$$

Taking the supremum over all $E \subseteq \mathcal{Y}$ gives

$$D_\infty(\mathcal{M}_H(q) \,\|\, \mathcal{M}_O(q)) \le \frac{\varepsilon}{b}, \qquad D_\infty(\mathcal{M}_O(q) \,\|\, \mathcal{M}_H(q)) \le \frac{\varepsilon}{b}.$$

Applying Lemma B.3 yields the stated inequalities:

$$D(\mathcal{M}_H(q) \,\|\, \mathcal{M}_O(q)) \le \frac{\varepsilon}{b} \cdot \big(e^{\varepsilon/b} - 1\big), \qquad D(\mathcal{M}_O(q) \,\|\, \mathcal{M}_H(q)) \le \frac{\varepsilon}{b} \cdot \big(e^{\varepsilon/b} - 1\big).$$

$\square$

### B.2.4 Proof of theorem 4.5

**Theorem 4.5.** *Let $S \in \text{Supp}(\mathcal{D})$ be a sample, and let $\mathcal{A}$ be a fixed analyst. Consider two $k$-round interactions with $\mathcal{A}$: one with the hybrid mechanism $\mathcal{M}_H$, and one with the oracle mechanism $\mathcal{M}_O$. For any $\rho > 0$, define $\varepsilon^* = \sqrt{2k\ln\left(\frac{1}{\rho}\right)}\frac{\varepsilon}{b} + k\frac{\varepsilon}{b}\left(e^{\varepsilon/b} - 1\right)$. Then*

$$\Pr_{t \leftarrow \mathcal{M}_H}\left[\ln\frac{\Pr_{\mathcal{M}_H}(T = t)}{\Pr_{\mathcal{M}_O}(T = t)} > \varepsilon^*\right] \leq \rho, \quad \text{and} \quad \Pr_{t \leftarrow \mathcal{M}_O}\left[\ln\frac{\Pr_{\mathcal{M}_O}(T = t)}{\Pr_{\mathcal{M}_H}(T = t)} > \varepsilon^*\right] \leq \rho$$

*Proof of theorem 4.5.* Fix an analyst $\mathcal{A}$ and a sample $S$. To show that

$$\Pr_{t \leftarrow \mathcal{M}_H}\left[\ln\frac{\Pr_{\mathcal{M}_H}(T = t)}{\Pr_{\mathcal{M}_O}(T = t)} > \varepsilon^*\right] \leq \rho$$

We begin by decomposing the log-likelihood ratio over the $k$ rounds:

$$\ln\frac{\Pr_{\mathcal{M}_H}[T = t]}{\Pr_{\mathcal{M}_O}[T = t]} = \sum_{i=1}^{k}\ln\frac{\Pr_{\mathcal{M}_H}[T_i = t_i \mid T_{i-1} = t_{i-1}]}{\Pr_{\mathcal{M}_O}[T_i = t_i \mid T_{i-1} = t_{i-1}]}$$

Since the analyst is assumed to be deterministic, the query in the $i$-th round is fully determined by the history up to that round. Therefore for any mechanism $\mathcal{M}$ and for any $t = (q_1, a_1, \ldots, q_k, a_k) \in \mathcal{T}_{\mathcal{A}}$, it holds that:

$$\Pr_M[T_i = t_i \mid T_{i-1} = t_{i-1}] = \Pr[Q_i = q_i \mid T_{i-1} = t_{i-1}] \cdot \Pr(\mathcal{M}(q_i) = a_i) = 1 \cdot \Pr(\mathcal{M}(q_i) = a_i)$$

Next, we define the random variable, for any $t = (q_1, a_1, \ldots, q_k, a_k) \in \mathcal{T}_{\mathcal{A}}$:

$$C_i(t_i) = \ln\frac{\Pr_{\mathcal{M}_H}[T_i = t_i \mid T_{i-1} = t_{i-1}]}{\Pr_{\mathcal{M}_O}[T_i = t_i \mid T_{i-1} = t_{i-1}]} = \ln\frac{\Pr(\mathcal{M}_H(q_i) = a_i)}{\Pr(\mathcal{M}_O(q_i) = a_i)}.$$

Then

$$\ln\frac{\Pr_{\mathcal{M}_H}(T = t)}{\Pr_{\mathcal{M}_O}(T = t)} = \sum_{i=1}^{k}C_i(t_i).$$

We want to apply Azuma–Hoeffding's inequality B.4 to the sequence $C_1, \ldots, C_k$, to show that for any $\rho > 0$

$$\Pr_{t \sim \mathcal{M}_H}\left[\sum_{i=1}^{k}C_i(t_i) > \sqrt{2k\ln\left(\frac{1}{\rho}\right)}\frac{\varepsilon}{b} + k\frac{\varepsilon}{b}(e^{\varepsilon/b} - 1)\right] \leq \rho$$

which implies that

$$\Pr_{t \leftarrow \mathcal{M}_H}\left[\ln\frac{\Pr_{\mathcal{M}_H}(T = t)}{\Pr_{\mathcal{M}_O}(T = t)} > \varepsilon^*\right] \leq \rho$$

To apply Azuma–Hoeffding's inequality (Lemma B.4) to the sequence $C_1, \ldots, C_k$, it suffices to verify the following conditions for all $i \in \{1, \ldots, k\}$:

1. $\Pr_{t \sim \mathcal{M}_H}\left(|C_i(t_i)| \leq \frac{\varepsilon}{b}\right) = 1$.

2. For any $c_1, \ldots, c_{i-1} \in \text{Supp}(C_1, \ldots, C_{i-1})$:

$$\mathbb{E}[C_i \mid C_1 = c_1, \ldots, C_{i-1} = c_{i-1}] \leq \frac{\varepsilon}{b} \cdot (e^{\varepsilon/b} - 1),$$

We now verify each item in turn. Fix any transcript $t = (q_1, a_1, \ldots, q_k, a_k) \in \mathcal{T}_{\mathcal{A}}$.

**Verification of 1.** From Lemma 4.4, for every round $i$, we have

$$C_i(t_i) = \ln\frac{\Pr[\mathcal{M}_H(q_i) = a_i]}{\Pr[\mathcal{M}_O(q_i) = a_i]} \leq \varepsilon/b, \qquad -C_i(t_i) = \ln\frac{\Pr[\mathcal{M}_O(q_i) = a_i]}{\Pr[\mathcal{M}_H(q_i) = a_i]} \leq \varepsilon/b.$$

Thus for any $t \in \mathcal{T}_{\mathcal{A}}$:

$$|C_i(t_i)| \leq \frac{\varepsilon}{b}.$$

**Verification of 2.** We begin by bounding the conditional expectation of $C_i$ given any fixed transcript prefix $t_{i-1} \in \mathcal{T}_\mathcal{A}$. Since the analyst is deterministic, fixing $t_{i-1}$ determines the query $q_i$, and the randomness in round $i$ lies only in the mechanism's response.

$$\mathbb{E}\big[C_i(T_i) \mid T_{i-1} = t_{i-1}\big] = \sum_{t_i' \in \mathcal{T}_\mathcal{A}} \Pr_{\mathcal{M}_H}\big[T_i = t_i' \big| T_{i-1} = t_{i-1}\big] \cdot C_i(t_i')$$

$$= \sum_{t_i' \in \mathcal{T}_\mathcal{A}} \Pr_{\mathcal{M}_H}\big[T_i = t_i' \big| T_{i-1} = t_{i-1}\big] \cdot \ln \frac{\Pr_{\mathcal{M}_H}\big[T_i = t_i' \big| T_{i-1} = t_{i-1}'\big]}{\Pr_{\mathcal{M}_O}\big[T_i = t_i' \big| T_{i-1} = t_{i-1}'\big]}$$

$$= \sum_{a \in \mathcal{Y}} \Pr[\mathcal{M}_H(q_i) = a] \cdot \ln \frac{\Pr[\mathcal{M}_H(q_i) = a]}{\Pr[\mathcal{M}_O(q_i) = a]}$$

$$= D\big(\mathcal{M}_H(q_i) \,\|\, \mathcal{M}_O(q_i)\big) \leq \frac{\varepsilon}{b}\big(e^{\varepsilon/b} - 1\big),$$

where the third equality uses the fact that $t_{i-1}' = t_{i-1}$, since $t_i' \sim \Pr_{\mathcal{M}_H}[\cdot \mid T_{i-1} = t_{i-1}]$, and the final inequality follows from Lemma B.8.

Note that $C_1, \ldots, C_k$ are deterministic functions of the transcript, meaning that the transcript prefix $t_{i-1}$ fully determines the values $c_1, \ldots, c_{i-1}$. Hence by showing the above bound holds for any transcript prefix in the support $\mathcal{T}_\mathcal{A}$, it implies the desired conditional expectation also holds for any $c_1, \ldots, c_{i-1} \in \mathrm{Supp}(C_1, \ldots, C_{i-1})$. Therefore:

$$\mathbb{E}[C_i \mid C_1 = c_1, \ldots, C_{i-1} = c_{i-1}] \leq \frac{\varepsilon}{b} \cdot (e^{\varepsilon/b} - 1).$$

**Conclusion.** Since both items 1 and 2 hold, Azuma–Hoeffding's inequality implies that for any $\rho > 0$,

$$\Pr_{t \sim \mathcal{M}_H}\Big[\sum_{i=1}^{k} C_i(t_i) > \sqrt{2k \ln\big(\tfrac{1}{\rho}\big)} \frac{\varepsilon}{b} + k \frac{\varepsilon}{b}(e^{\varepsilon/b} - 1)\Big] \leq \rho,$$

as claimed.

A symmetric argument applies to the reversed log-likelihood ratio, by repeating the analysis with the roles of $\mathcal{M}_H$ and $\mathcal{M}_O$ swapped. Hence, for any $t \in \mathcal{T}_\mathcal{A}$, we obtain

$$\Pr_{t \leftarrow \mathcal{M}_H}\left[\ln \frac{\Pr_{\mathcal{M}_H}(T = t)}{\Pr_{\mathcal{M}_O}(T = t)} > \varepsilon^*\right] \leq \rho, \quad \text{and} \quad \Pr_{t \leftarrow \mathcal{M}_O}\left[\ln \frac{\Pr_{\mathcal{M}_O}(T = t)}{\Pr_{\mathcal{M}_H}(T = t)} > \varepsilon^*\right] \leq \rho$$

$\square$

## B.3 Proof of corollary 4.6

**Corollary 4.6.** *Let $\mathcal{A}$ be a fixed analyst, $S \in \mathrm{Supp}(\mathcal{D})$ a sample, and let $\mathcal{E}$ be any event that can arise in the interaction with the analyst. For any $\rho > 0$, define $\varepsilon^*$ as in Theorem 4.5. Then*
$$e^{-\varepsilon^*}\left(\Pr_{\mathcal{M}_O}[\mathcal{E}] - \rho\right) \leq \Pr_{\mathcal{M}_H}[\mathcal{E}] \leq e^{\varepsilon^*} \Pr_{\mathcal{M}_O}[\mathcal{E}] + \rho$$

*Proof.* Let $\mathcal{B} = \left\{t \in \mathcal{T}_\mathcal{A} : \ln \frac{\Pr_{\mathcal{M}_H}(T=t)}{\Pr_{\mathcal{M}_O}(T=t)} > \varepsilon^*\right\}$ be the "bad" event where the likelihood-ratio bound fails. By Theorem 4.5 $\Pr_{\mathcal{M}_H}[\mathcal{B}]$ is at most $\rho$. Hence

$$\Pr_{\mathcal{M}_H}[\mathcal{E}] = \Pr_{\mathcal{M}_H}[\mathcal{E} \cap \mathcal{B}] + \Pr_{\mathcal{M}_H}[\mathcal{E} \cap \mathcal{B}^c] \leq \rho + e^{\varepsilon^*} \Pr_{\mathcal{M}_O}[\mathcal{E} \cap \mathcal{B}^c] \leq \rho + e^{\varepsilon^*} \Pr_{\mathcal{M}_O}[\mathcal{E}].$$

Exchanging roles of $\mathcal{M}_H$ and $\mathcal{M}_O$ yields the lower bound $\Pr_{\mathcal{M}_H}[\mathcal{E}] \geq e^{-\varepsilon^*}(\Pr_{\mathcal{M}_O}[\mathcal{E}] - \rho)$. $\square$

## B.4 Proof of lemma 4.7

**Lemma 4.7.** *Fix an analyst $\mathcal{A}$, a noise scale $b$, and let $\mathcal{G}$ be the set of $\varepsilon$-good sample–transcript pairs. Then, for every measurable subset $\mathcal{E} \subseteq \mathcal{G}$, $\Pr_{\mathcal{M}_S}[\mathcal{E}] = \Pr_{\mathcal{M}_H}[\mathcal{E}]$.*

*Proof of lemma 4.7.* Run both mechanisms by first drawing the sample $S \sim \mathcal{D}$ and then drawing $k$ independent Laplace noises $\eta_1, \ldots, \eta_k \sim Laplace(0, b)$. These draws fix all randomness in the interaction. On any transcript $t$ in the event $\mathcal{E}$, every query $q_i$ satisfies $|q_i(S) - q_i(\mathcal{D})| \leq \varepsilon$, so by definition, the hybrid mechanism never switches to the oracle mode. Hence for every draw $(S, \eta_1, \ldots, \eta_k)$ that yields $t$, both $\mathcal{M}_S$ and $\mathcal{M}_H$ produce the same $t$. Since the joint distribution over $(S, \eta_1, \ldots, \eta_k)$ is identical in both mechanisms, the probability of observing any $t \in \mathcal{E}$ is the same. $\square$

## B.5 Proof of lemma 4.8

**Lemma 4.8.** *Let $S \sim \mathcal{D}$ and consider a $k$-round interaction between an analyst $\mathcal{A}$ and the oracle mechanism, producing the transcript $t$. Define the failure probability of any of the Laplace noises exceeding $\alpha$ as $\zeta = 1 - \Pr[|\eta_1'| \leq \alpha]^k, \quad for \ \eta_1' \sim \mathrm{Laplace}(0, b)$. Then,*

$$\Pr_{S \sim \mathcal{D},\, t \sim Pr_{\mathcal{M}_O}} \left[ (S, t) \text{ is } \varepsilon\text{-good and } t \text{ is } \alpha\text{-accurate} \right] \ \geq \ 1 - k\gamma - \zeta.$$

*Proof of lemma 4.8.* Since the oracle mechanism operates independently of the sample, the queries are chosen independently of the sample. By the definition of $(\varepsilon, \gamma)$-concentration and applying a union bound over all $k$ queries, the probability that the sample-transcript pair is $\varepsilon$-good is at least $1 - k \cdot \gamma$. Additionally the probability that for all $k$ rounds the oracle's response is within $\alpha$ of the true mean is $1 - \zeta$, where $\zeta$ represents the failure probability due to the added Laplace noises. Combining these bounds with a union bound yields the desired result. $\square$

## B.6 Proof of theorem 4.9

**Theorem 4.9.** *Let $\mathcal{A}$ be an analyst, and $\mathcal{M}_S$ the real-world Laplace mechanism interacting with $\mathcal{A}$ over $k$ rounds. For $\alpha > 0$ and $\rho > 0$, define $\varepsilon^*$ as in theorem 4.5, and let $\zeta$ be as in lemma 4.8. Then, the probability that the real-world mechanism produces an $\alpha$-accurate transcript satisfies*

$$\Pr_{\mathcal{M}_S} \left[ t = (q_1, a_1, \ldots, q_k, a_k) \ : \ \forall i \ |a_i - q_i(\mathcal{D})| \leq \alpha \right] \ \geq \ e^{-\varepsilon^*} \left( 1 - k\gamma - \zeta - \rho \right).$$

*Proof of theorem 4.9.* Let $\mathcal{E}$ denote the event that a sample–transcript pair $(S, t)$ is both $\varepsilon$-good and $\alpha$-accurate. By Lemma 4.8, we have $\Pr_{\mathcal{M}_O}[\mathcal{E}] \geq 1 - k\gamma - \zeta$. Applying Lemma 4.6, the probability of $\mathcal{E}$ under the hybrid mechanism is $\Pr_{\mathcal{M}_H}[\mathcal{E}] \geq e^{-\varepsilon^*}(\Pr_{\mathcal{M}_O}[\mathcal{E}] - \rho)$. By Lemma 4.7, we know the probabilities for $\varepsilon$-good pairs are identical for the real-world and hybrid mechanisms, so $\Pr_{\mathcal{M}_S}[\mathcal{E}] = \Pr_{\mathcal{M}_H}[\mathcal{E}]$. Since $\mathcal{E}$ is a subevent of the event that the transcript is $\alpha$-accurate, we conclude that the probability of an $\alpha$-accurate transcript is at least $e^{-\varepsilon^*}(1 - k\gamma - \zeta - \rho)$. $\square$

## B.7 Detailed derivation of the query bound

**Theorem 4.10.** *Let $\mathcal{A}$ be any analyst issuing $k$ adaptive $(\varepsilon, \gamma)$-concentrated queries, and fix an accuracy parameter $\alpha > 0$ and failure probability $\beta > 0$. Then the Laplace mechanism can achieve $(\alpha, \beta)$-accuracy over all $k$ queries provided $k = O\left( \min\left\{ \frac{\beta}{\gamma}, \ \beta \varepsilon^{-2}, \ \frac{\alpha^2 \beta^2}{\varepsilon^2 [\ln(1/\varepsilon)]^2 \ln(1/\beta)} \right\} \right).$*

*Full derivation of the bounds in theorem 4.10.* Theorem 4.9 implies that for any fixed $\alpha > 0$ and number of queries $k$, the real-world mechanism satisfies $(\alpha, \beta)$-accuracy provided

$$\Pr \left[ \forall i : \ \left| q_i(S) + \eta_i - q_i(\mathcal{D}) \right| \leq \alpha \right] \ \geq \ e^{-\varepsilon^*} \cdot \left( 1 - k\gamma - \zeta - \rho \right) \ \geq \ 1 - \beta.$$

As in the proof of Theorem 4.10, we set $b = \frac{\alpha}{2 \ln(1/\varepsilon)}$. The overall failure probability depends on four terms: $\rho, k\gamma, \zeta,$ and $\varepsilon^*$, each of which is bounded by $\beta/4$. Under these conditions, the total success probability is approximated by

$$e^{-\varepsilon^*} \cdot \left( 1 - k\gamma - \zeta - \rho \right) \ \approx \ e^{-\beta/4} \cdot e^{-3\beta/4} = e^{-\beta} \ \approx \ 1 - \beta,$$

Concretely:

**(i) Concentration failure.**

$$k\,\gamma \;\le\; \frac{\beta}{4} \quad \Longrightarrow \quad k \;\le\; \frac{\beta}{4\,\gamma} = O\!\left(\tfrac{\beta}{\gamma}\right).$$

**(ii) Noise-exceedance failure.**  Recall $\zeta = 1 - (1 - e^{-\alpha/b})^k \le k\,e^{-\alpha/b}$. Requiring

$$k\,e^{-\alpha/b} \;\le\; \frac{\beta}{4} \quad \Longrightarrow \quad k \;\le\; \frac{\beta}{4}\,e^{\alpha/b} = O\!\left(\beta\,\varepsilon^{-2}\right),$$

**(iii) Likelihood-ratio failure.**  The term $\varepsilon^* = \sqrt{2k\ln(1/\rho)}\,\dfrac{\varepsilon}{b} + k\,\dfrac{\varepsilon}{b}(e^{\varepsilon/b} - 1)$ must satisfy $\varepsilon^* \le \beta/4$. With $\rho = \beta/4$ and using the bound $e^{\varepsilon/b} - 1 \le 2\varepsilon/b$ for $\tfrac{\varepsilon}{b} \in (0,1)$, we obtain

$$\sqrt{2k\,\ln(4/\beta)}\,\frac{\varepsilon}{b} + 2k\left(\frac{\varepsilon}{b}\right)^2 \;\le\; \frac{\beta}{4}.$$

Substitute $b = \frac{\alpha}{2\ln(1/\varepsilon)}$, so $\frac{\varepsilon}{b} = 2\varepsilon\,\frac{\ln(1/\varepsilon)}{\alpha}$. This yields

$$k = O\!\left(\frac{\alpha^2\,\beta^2}{\varepsilon^2\,[\ln(1/\varepsilon)]^2\,\ln(1/\beta)}\right).$$

**Conclusion.**  Taking the minimum over the three derived bounds on $k$ completes the proof of Theorem 4.10

$\square$

