# OpenReview forum: "Tight Bounds for Answering Adaptively Chosen Concentrated Queries"
_NeurIPS.cc/2025/Conference — NeurIPS 2025 poster_

### Official Review · Reviewer_oH1p · 2025-06-27

**Clarity:** 4
**Significance:** 3
**Originality:** 3
**Rating:** 5
**Confidence:** 4

**Summary:**

The authors study the limitation of adaptively answering $(\alpha,\beta)$-statistically accurate queries. Their main (negative) result shows, given the underlying distribution is $(\epsilon,\gamma)$-concentrate, no mechanism can adaptively answer more than $1/\gamma \wedge 1/\epsilon^2\log(1/\epsilon\gamma)$ queries. The authors also provide a positive result that roughly matches the above bound up to polylog factors with respect to $\epsilon$ and $\gamma$.

**Questions:**

1. What would be the canonical distributions (beyond i.i.d.) that satisfy the concentration property? e.g. maybe finite-state Markov chains with a concentrated stationary distribution, or some martingales?

2. page 2, line 44, $\sum_{x\in T} q(x) \rightarrow \sum_{x\in T} q_j(x)$?

**Ethical Concerns:**

["NO or VERY MINOR ethics concerns only"]

**Final Justification:**

The authors have answered my questions. I'll keep my (already high) score.

**Limitations:**

yes

**Quality:**

4

**Strengths And Weaknesses:**

Strengths:

> The results appear to be quite significant. It is potentially fundamental, given the natural formulation of the problem. Although I am not an expert in differential privacy, the authors’ findings still look very interesting to me. Thus it is likely to attract interest well beyond the privacy community.

> The paper is enjoyable to read. Despite not having a background in privacy, I found the problem setup very clear, the significance of the results compelling, and the proof techniques both accessible and well-explained. The proof (based on what I read) is also well-structured and clear.

> The lower bound proof is very clever. The simplified ($1/\gamma$) version is quite simple but already interesting. The general version is pretty neat.

Weaknesses:

I do not see any significant weaknesses from my perspective. However, I believe the paper could be more impactful if the authors elaborated further on the potential applications of these adaptive queries, as there may be many relevant use cases.

---

> ### Author Rebuttal · Authors · 2025-07-30
>
> Thank you for your thoughtful review. We will correct the typo you identified and expand on the motivation for considering adaptive queries in the next revision of our work.
>
> **> What would be the canonical distributions (beyond i.i.d.) that satisfy the concentration property?  e.g. maybe finite-state Markov chains with a concentrated stationary distribution, or some martingales?**
>
> Any canonical distribution that is known to satisfy a concentration bound analogous to Hoeffding’s inequality for i.i.d. samples (e.g., Azuma’s inequality for martingales with bounded differences) would indeed satisfy the concentration property. Exploring such distributions, particularly those that naturally arise in real-world sampling scenarios, is a promising direction for future work. Many of these distributional structures provide stronger concentration guarantees than the general assumption used in this paper, while still relaxing the very strict i.i.d. assumption.

---

> > ### Comment · Reviewer_oH1p · 2025-08-04
> >
> > Thanks for the response! I'll keep my score.

---

### Official Review · Reviewer_LHqT · 2025-07-03

**Clarity:** 4
**Significance:** 3
**Originality:** 3
**Rating:** 5
**Confidence:** 3

**Summary:**

This work considers an adaptive data analysis (ADA) game between an adversarial analyst $A$ and a mechanism $M$. In particular, $M$ is provided with a set $S$ of $n$ samples from a domain $X$ drawn from a distribution $D$. Then over $k$ rounds, $A$ submits queries $q_j  \colon X \to [0,1]$, and $M$ must return an answer $a_j \in \mathbb{R}$. The mechanism’s goal is to produce answers ($a_j$’s) that are close to the average value of the query over the true distribution, but the catch is that $M$’s only knowledge of the distribution is through the sample $S$ it holds. The question is, how many queries can $M$ answer accurately? (Defined by an error parameter $\alpha$ and a confidence parameter $\beta$.) As the number of samples, $n$, increases, $M$ is able to answer more queries.

This problem is well-studied under the assumption that the $n$ samples are obtained independently, and here it is known that $\Theta(n^2)$ queries can be answered. If this independence assumption is removed without changing anything else about the problem, then this problem suddenly becomes impossible, as the authors point out with a simple example. One way forward is to then restrict the types of queries that the analyst is allowed to make. This work continues in the footsteps of Bassily-Freund 2016, who considered this question when the analyst is only allowed to make queries that are well-concentrated around their mean.

It was known due to prior work that $O(n)$ queries can be answered in this setting, quadratically fewer than when samples are independent. This work shows that this quadratic gap is necessary for all noise addition mechanisms, which are a natural special class. On the other hand, they provide a simpler analysis of the mechanism due to Bassily-Freund using techniques from differential privacy.

**Questions:**

Suggestions / questions:

It took me a bit of time to parse and understand the definition of the analyst-mechanism game on page 1, although I was able to do so after some thought. However, it feels like the presentation could be improved a bit by providing a natural example and explaining what the different objects ($A,M,D$ queries, etc.) represent in the example.

Instead of limiting the types of queries made by the analyst, what if one relaxes the independence assumption a bit? Are there known bounds for this problem quantified in terms of “how independent” the samples are guaranteed to be?

Minor typos and comments:

Line 44: Should $q_j$ be just $q$ in this expression?

Line 49: “of sample” should be “of the sample”?

Paragraph labeled “1” on page 2 contains a few typos: “chose” -> “choose”, “to to” -> “to.

Line 112: “adhere the” -> “adhere to the”?

**Ethical Concerns:**

["NO or VERY MINOR ethics concerns only"]

**Final Justification:**

I believe this is a well-written paper with nice results which is above the bar for NeurIPS. I am keeping my score a 5.

**Limitations:**

yes

**Quality:**

3

**Strengths And Weaknesses:**

Strengths: This paper is nicely written in my opinion. I also appreciate the theme of the paper which is to remove a rather strict assumption (sample independence) from a natural statistical algorithmic problem. The paper makes nice contributions in this direction by proving a lower bound for a natural class of mechanisms and simplifying existing upper bounds by tying in established techniques, which I believe is valuable.

Weaknesses: The fact that the lower bound only holds for a certain special case of mechanisms could be considered a weakness.

Overall, I think the paper merits acceptance.

---

> ### Author Rebuttal · Authors · 2025-07-30
>
> Thank you for your constructive review and helpful suggestions.
>
> **> The fact that the lower bound only holds for a certain special case of mechanisms could be considered a weakness.**
>
> Our lower bound is indeed limited to noise-addition mechanisms. At the same time, noise-addition mechanisms remain a broad and important class, used both on their own and also as key building blocks in many ADA mechanisms. Our lower bound also applies to mechanisms that answer queries by evaluating them on random subsamples (this could be viewed as a form of a mean-zero noise addition mechanism; we will clarify this in the next revision). Thus, our lower bound shows that all known (efficient) techniques for answering a super-linear number of queries in the i.i.d. setting provably fail when correlations are present in the data. As we mentioned at the end of Section 1.1.1, our work steers future work to either impose additional structural assumptions on the problem or introduce fundamentally new algorithmic techniques.
>
> **> It feels like the presentation could be improved a bit by providing a natural example and explaining what the different objects represent in the example.**
>
> Thank you for this suggestion; we will try to make the introduction of the general setting more accessible and will also correct the typos you noted in the next revision of our work.
>
> **> Instead of limiting the types of queries made by the analyst, what if one relaxes the independence assumption a bit? Are there known bounds for this problem quantified in terms of "how independent" the samples are guaranteed to be?**
>
> This is indeed a promising direction for future work. One recent work (Kontorovich et al., 2022) addresses this setting, where the bound depends on a measure of the distance between the underlying distribution and a product distribution. To the best of our knowledge, no other results directly address these questions. Further investigation of this problem, particularly for distributional structures that naturally arise in sampling scenarios and have practical applications, would be a valuable direction for future research.

---

> > ### Comment · Reviewer_LHqT · 2025-08-01
> > **Rebuttal response**
> >
> > Thank you very much for your thoughtful responses to my questions!
> >
> > **Regarding the lower bound being limited to noise-addition mechanisms.** I appreciate your response to this point, especially the observation that your work "steers future work to either impose additional structural assumptions on the problem or introduce fundamentally new algorithmic techniques". After reflecting on it more and reading your response I actually don't think this should be seen as a weakness.
> >
> > **Regarding relaxing the independence assumption.** Thank you for pointing me to the work of Kontorovich et al. 2022. This gives useful context for my question.
> >
> > Thank you again for addressing my concerns! I have no further questions.

---

### Official Review · Reviewer_Whdi · 2025-07-03

**Clarity:** 3
**Significance:** 3
**Originality:** 3
**Rating:** 5
**Confidence:** 3

**Summary:**

This paper studies adaptive data analysis beyond the iid setting. Given a dataset drawn (perhaps non-iid) from a distribution $\cal{D}$ over datasets, the goal is to answer as many adaptively chosen statistical queries as possible, up to some error tolerance (relative to the expectation over $\cal{D}$) and failure probability.

In general, in the non-iid setting, there is no hope that this is possible, even if the queries are non-adaptive. This paper studies the restricted setting where the queries are enforced to be those which exhibit concentration in the non-adaptive setting. A query is $(\epsilon, \gamma)$-concentrated if the probability that the query deviates from its expectation by more $\epsilon$ is at most $\gamma$. In the non-adaptive setting, $1/\gamma$ of these queries could be answered all up to error $\epsilon$ with probability $3/4$ by a standard union bound. The main question addressed in this paper is what is the cost of adaptivity in the concentrated query setting?

Prior work of Bassily and Freund gave an algorithm allowing for $\tilde{O}(1/\epsilon^2)$ queries for $(\epsilon, n^{-10})$-concentrated queries. This allows for $\tilde{O}(n)$ queries which exhibit standard $1/\sqrt{n}$ error bounds, which is significantly less than results for adaptive queries in the iid setting which allow for quadratically many queries. The authors show that this is inherent to noise-additive mechanism for this problem (which include Bassily and Freund's result). Such mechanisms cannot accurately answer more than $\tilde{\Omega}(1/\epsilon^2)$ queries. On the upper bounds side, the authors give a simplified upper bound achieving this rate (up to log factors) via differential privacy.

**Questions:**

What is known about algorithms which do not fall into the class of "noise-additive" algorithms?

Are there natural weaker or incomparable restrictions of the allowable queries which may allow stronger positive results?

**Ethical Concerns:**

["NO or VERY MINOR ethics concerns only"]

**Final Justification:**

I maintain my positive score for the reasons given in the review.

**Limitations:**

Yes.

**Quality:**

3

**Strengths And Weaknesses:**

This paper forms a nice picture about how the restriction of correlated queries can be used to adaptively answer statistical queries. The picture is essentially complete under the restriction of noise-additive algorithms, with new lower bounds showing that the concentrated queries setting does not allow for the same performance as in the i.i.d. case.

This result can help to guide research towards other relaxations of the i.i.d. assumption that may allow for stronger adaptivity results.

## Minor Comments

Line 116: of -> or

In Theorem 1.4 and 3.1, I would use big-O rather than big-$\Omega$ for $k$ as the result is giving an upper bound on the smallest $k$ required to break the mechanism. (Though I understand the usage of these two notations is often somewhat subjective.)

---

> ### Author Rebuttal · Authors · 2025-07-30
>
> Thank you for your helpful suggestions and for pointing out the typo; we will address these in the next revision of our work.
>
> **> What is known about algorithms which do not fall into the class of "noise-additive" algorithms?**
>
> Currently there are two known techniques for (efficiently) answering adaptive queries when correlations are present in the data:
>
> **(1) Noise addition.** With correlations, all existing noise-addition mechanisms support at most $O(n)$ queries. We show that this is a barrier one cannot cross, despite the fact that, in the i.i.d. setting, noise addition supports up to $O(n^2)$ queries.
>
> **(2) Transcript compression.** Another technique for efficiently answering $O(n)$ queries when correlations are present is based on "transcript compression" (a.k.a. "Bounded description length"). But this technique only answers $O(n)$ queries even in the i.i.d. setting, and so one should not expect it to support a super-linear number of queries in the correlated case.
>
> Our lower bound also applies to mechanisms that answer queries by evaluating them on random subsamples (this could be viewed as a form of a mean-zero noise addition mechanism; we will clarify this in the next revision). Together, our lower bound shows that all known (efficient) techniques for answering a super-linear number of queries in the iid setting provably fail when correlations are present in the data.
>
> **> Are there natural weaker or incomparable restrictions of the allowable queries which may allow stronger positive results?**
>
> One natural direction for future work is to consider queries satisfying stronger concentration constraints, such as sub-gaussian queries. Another promising direction—potentially more applicable to natural scenarios—would be to impose stronger structural constraints on the underlying data distribution, while still relaxing the strict i.i.d. assumption.

---

> > ### Comment · Reviewer_Whdi · 2025-08-04
> > **Response to Author Rebuttal**
> >
> > Thanks for addressing my questions.

---

### Official Review · Reviewer_MjLV · 2025-07-03

**Clarity:** 4
**Significance:** 3
**Originality:** 4
**Rating:** 5
**Confidence:** 3

**Summary:**

This paper studies the problem of answering adaptive queries under the restriction that the queries are $(\varepsilon, \gamma)$-concentrated around its means. This concentration assumption is needed to enable adaptive analysis on dataset with correlations (beyond the typical i.i.d setting). They provide query bounds that are tight up to logarithmic factors when the query answer provider is restricted to noise addition mechanisms.

Their impossibility result is new and implies an inherent gap between i.i.d. setting and setting with correlations (when restricted to noise addition mechanisms). While the newly provided noise addition mechanism is only a logarithmic improvement over the prior work, the analysis, combining ideas from differential privacy, is simplified compared to previous works.

**Questions:**

Suggestion and typos:
1. I am surprised to see the $O(n)$ v.s. $O(n^2)$ gap persists even if the queries are tightly concentrated as Hoeffding bound. Is there a intuitive explanation to why Hoeffiding bound is insufficient to support more queries in correlation setting. What other properties of i.i.d. setting are exploited to obtain the $O(n^2)$ query result.
2. Maybe give one simple example to show why adaptive queries are hard to answer, in the spirit of the example given in 56-62? What can the adaptive adversaries do to make the query answering hard?
3. On my first read-through, I was confused by the expectation on Line 44 as it is a bit uncommon to treat a tuple or an entire dataset as "a fresh sample": $T \leftarrow D$. Perhaps it would read better if you repeat here that $T$ is a tuple or an entire dataset.
4. On Line 90, should $q(T)$ be the average of the tuple like what you wrote on Line 44 or on the same line? It doesn't make sense to me otherwise.
5. In the standard upper and lower bounds kind of language, Theorem 1.4 and Theorem 1.5, is quite the opposite -- the hard instance (typically called lower bound) has larger values than what is reflected in the algorithm (typically called upper bound). Perhaps, it would be easier for folks like me to read if you add a note somewhere nearby explaining that query bounds are indeed the opposite of the time complexity bounds?
   On that note, maybe it’s worth drawing a 2D-graph depicting the feasible regimes and infeasible regimes here to kind of map out this $\min(1/\gamma, \ln(1/(\eps \gamma))/\eps^2)$ v.s. $\min(1/\gamma, 1/(\eps^2 (\ln(1/\eps))^2))$ query bounds?
6. On Line 133, there is a typo, $\delta$ should be $\gamma$.
7. On Line 139, it reads "A simplified positive result". On first read-through, I thought this was a simplified result compared to a more involved and general result in your work. Maybe change this to “alternative” or “A positive result based on techniques from differential privacy” or "A simplified positive result compared to *Bassily and Freund [2016]"*?
8. I guess idea of Algorithm 1 is based on the idea from Line 226-232 -- identify the one query that can fail the mechanism? Probably it will the reading easier if the authors add the connection to this simpler starting example on Line 251 (attack overview).
9. On Line 253: why focus on $\mathcal{X}_1$, can we also make the same argument on $\mathcal{X}_2$?
10. From the presentation of the Algorithm 1 (or the description from Line 251-263), it is can be unclear that the Bernoulli chosen is fixed for the same $x$, i.e., it can be confused with the case where each query is simply randomly picking a fresh Bernoulli sample with parameter $p_t$. While it is clear once you take a look at the proof in the Appendix, I would suggest the authors to rewrite this part to make it clear that $q_t(x)$ is determinisic once you fix the random seeds of this Bernoulli (or make $q_t$ take the random seeds as one of the inputs).
11. On Line 337, there is a typo double comma: ", ,".

**Ethical Concerns:**

["NO or VERY MINOR ethics concerns only"]

**Final Justification:**

I maintain my ratings after the discussion.

**Limitations:**

Yes.

**Paper Formatting Concerns:**

No concern.

**Quality:**

4

**Strengths And Weaknesses:**

Strengths
1. This paper is well written and clear in its presentation. I appreciate the example given in the introduction in addition to the informal proof techniques, which are also very helpful for understanding the proofs.
2. While I did not fully check proofs in the Appendix, the proofs and proof sketch on the main body seem solid.
3. Their result settles the problem (up to logarithmic facotrs) when the mechanism is restricted to adding noise.

Weaknesses:
   While the question considered is indeed quite interesting, I fail to see the motivation to restrict the setting to noise addition mechanism. Maybe the authors would like to give some more background on this aspect? Was this a popular or most promising method to study before this work?

---

> ### Author Rebuttal · Authors · 2025-07-30
>
> Thank you for your detailed and helpful review. We greatly appreciate the comprehensive suggestions and corrections regarding clarity, readability, and typos, and we will address them carefully in the revision.
>
> **> I fail to see the motivation to restrict the setting to noise addition mechanism. Maybe the authors would like to give some more background on this aspect? Was this a popular or most promising method to study before this work?**
>
> Noise-addition is a very popular approach, used both on its own and as a fundamental building block in many ADA algorithms. Our lower bound also applies to mechanisms that answer queries by evaluating them on random subsamples (this could be viewed as a form of a mean-zero noise addition mechanism; we will clarify this in the next revision). Thus, our lower bound shows that all known (efficient) techniques for answering a super-linear number of queries in the i.i.d. setting provably fail when correlations are present in the data. As we mentioned at the end of Section 1.1.1, our work steers future work to either impose additional structural assumptions on the problem or introduce fundamentally new algorithmic techniques.
>
> **> I am surprised to see the $O(n)$ v.s. $O(n^2)$ gap persists even if the queries are tightly concentrated as Hoeffding bound. Is there an intuitive explanation to why Hoeffiding bound is insufficient to support more queries in correlation setting? What other properties of i.i.d. setting are exploited to obtain the query result?**
>
> This is indeed a surprising result and one of the compelling insights of our paper. To gain intuition for this gap, consider first the simpler i.i.d. setting. A typical query in the i.i.d. scenario has a standard deviation on the order of $O(1/\sqrt(n))$. Thus, naively, to "hide" deviations in even a single query, one might expect to add noise of this magnitude. By leveraging advanced composition arguments from the literature of differential privacy, answering $k$ adaptive queries would therefore (naively) require noise on the order of $O(\sqrt(k)/\sqrt(n))$, limiting the number of queries $k$ to be linear in $n$. The key strength of differential privacy in the i.i.d. setting is that the magnitude of noise needed per query only needs to "hide" every single sample point, which translates to significantly less noise than the standard deviation. This is what enables answering a super-linear number of adaptive queries in the i.i.d. setting. However, our result shows that once correlations are introduced—even with queries that satisfy strong concentration properties— this advantage disappears. In other words, when correlations are present in the data, our results show that (in general) one must add noise of the order of the standard deviation, and that "hiding" every single sample point is insufficient.

---

> > ### Comment · Reviewer_MjLV · 2025-08-04
> >
> > Thank you for your response and insights!

---

### Official Review · Reviewer_GwEQ · 2025-07-08

**Clarity:** 3
**Significance:** 2
**Originality:** 2
**Rating:** 4
**Confidence:** 4

**Summary:**

The paper considers adaptive data analysis, in the case where the dataset observed by the mechanism _does not_ consist of independent samples - indeed arbitrary correlations are permitted, and relatively weak assumptions on the pair of the query/dataset distribution are inputed.

Concretely, as commonly studied in the area of differential privacy, a fruitful line of work starting with Dwork et al paper, shows that a mechanism given access to $n$ independent samples from a distribution $\mathcal{P}$ over $\mathcal{X}$, can provide an $0.1$ accurate answer to a sequence of $O(n^2)$ statistical queries; where a statistical query is specified my "analyst", as a function $q_i : \mathcal{X} \to [0,1]$, and the mechanism aims to provide answers $a_i$ s.t. $|\E_{x \sim \mathcal{P}} q_i(x) - a_i| < 0.1$. (Naive approach would use $O(1)$ fresh samples for each query, being able to answer only $O(n)$ adaptive queries of this form.)

In the long line of research on the adaptive data analysis, the assumption that the dataset is given by sampling $n$ independent samples from some distribution is prevalent. This paper, after Bassily and Freund investigates a dramatic weakening of this assupton.

Specifically, a query $q$ (for a distribution $\mathcal{D}$ over $m$ tuples in $\mathcal{X}$) is said to be $(\varepsilon, \gamma)$-concentrated, if $\Pr_{D \sim \mathcal{D}}(|q(D) - q(\mathcal{D})| > \varepsilon) < \gamma,$ where $q(D) := \frac{1}{m} \sum_{x \in D} q(x)$ and $q(\mathcal{D}) := \mathbb{E}_{D \sim \mathcal{D}} q(D)$. If a distribution $\mathcal{D}$ is a product distribution, any bounded query is $(O(\log n/\sqrt{n}), n^{10})$-concentrated, or more generally, it's $(\varepsilon, \exp(-\Omega(\varepsilon^2 n)))$-concentrated for every $\varepsilon$.

Bassily and Freud showed that there is a mechanism which could answer around $\tilde{O}(1/\varepsilon^2)$ adaptive queries as long as they are all $(\varepsilon, poly(\varepsilon))$-concentrated; specializing this to a setting where the database is formed by independent samples, this only recovers a simple $O(n)$ query budget (by using fresh samples for each query), as opposed to the improved $O(n^2)$ query budget from differential privacy.

In this paper, the authors provide a new, simpler analysis qualitatively matching (and slightly improving in lower order terms) the result of Bassily and Freud, as well as provide a new lower bound: for a large class of mechanisms that operates by adding independent noise to a sample average of each query, one cannot hope to answer significantly more than $O(1/\varepsilon^2)$ queries.

**Questions:**

The paper is generally quite well written. One striking gap is that throughout the paper, it seems that it is never formally stated what is supposed to mean that the lower bound rules out any mechanism that operated by adding "any noise". Looking at the proof in the appendix, we see that the assumptions used in the lower bound proof together are quite unrealistic (they do not capture any of the existing mechanisms), although it seems that this is only for the convenience in the proof, not any intrinsic issues, and the same ideas should provide a lower bound under more realistic noise model.

Specifically, as I understand it, the mechanism is assumed to provide answers of the form $a_t = q_t(D) + \eta_t$, s.t. $\eta_t$ are i.i.d. random variables from some distributions, with $\mathbb{E} \eta_t = 0$ (line 816). But in the paragraph on line 218 it is assumed that the outputs are clipped to an interval $[0,1]$ or "slightly extended" (some bound on the range is used later in the appendix, in line 826). Discussion about a noise having non-neglible chance exceeding "any reasonable range" seems quite imprecise.

**Ethical Concerns:**

["NO or VERY MINOR ethics concerns only"]

**Final Justification:**

I seem to be a bit less enthusiastic than other reviewers regarding the significance of this specific result; but in the end they do provide a neat, clean and short proof of a upper bound, together with a matching lower bound for a specific class of mechanism. I agree that the general direction of adaptive data analysis is significant, and the goal of understanding how the independence assumption can be relaxed is worth pursuing.

**Limitations:**

The lower bound applies only to a specific type of mechanisms, addition of independent mean-zero noise to each query. This lower bound is not as surprising, since the $(\varepsilon, \gamma)$-correlated queries assumption on the data/query distribution is extremely weak (compared to, say independent samples - but also much weaker than many other models of correlated samples). The weak assumptions for the data distribution is a pro for the upper bound, but the mechanism only slightly improves parameters over the original Bassily and Fruend algorithm.

**Paper Formatting Concerns:**

No concerns.

**Quality:**

3

**Strengths And Weaknesses:**

The paper clarifies completely the situation of adaptive data analysis under the very weak assumptions of Bassily and Freund -- provides a short and intuitive proof of correctness, leveraging ideas from differential privacy (which is standard now in the area - and as simpler and more intuitive, than the previous _ad hoc_ analysis), while simultanously proving a tight lower bound - under this specific weakning of the assumptions, any mechanism that just adds independent mean-zero noise cannot achieve any better trade-offs than proposed here.

The main two weaknesses are that the lower bound applies only for mechanisms that operate by adding independent noise to a sample average -- there is no specific reason why a mechanism needs to operate this way, so it would be much more interesting to see a more general lower bound (which seems likely, since the conditions are very weak).

Also, potentially some more explorations whether the $(\varepsilon, \gamma)$-concentration condition can be weakened would be interesting. As it is right now, this condition concerns a single point in the tail distribution of a random variable $q(D)$ for $D \sim \mathcal{D}$ -- the probability of it's deviation from mean exceeding $\varepsilon$ should be at most $\gamma$. When the dataset is given by i.i.d. samples, the random variable $q(D)$ is $1/\sqrt{n}$ subgaussian, i.e. it is $(\varepsilon, \exp(-\varepsilon^2 n))$-concentrated _for every_ $\varepsilon$. Would a strenghtening of the conditions to one of this form be enough to obtain an improved upper bound? Or can the lower bound be modified to rule out existance of noise-adding algorithm under this more stringent requirement?

---

> ### Author Rebuttal · Authors · 2025-07-30
>
> Thank you very much for your thorough review. We appreciate your thoughtful comments and will try to address your concerns in turn.
>
> **> The lower bound applies only for mechanisms that operate by adding independent noise to a sample average -- there is no specific reason why a mechanism needs to operate this way, so it would be much more interesting to see a more general lower bound.**
>
> Our lower bound is indeed limited to noise-addition mechanisms. At the same time, noise-addition mechanisms remain a broad and important class, used both on their own and also as key building blocks in many ADA mechanisms. Our lower bound also applies to mechanisms that answer queries by evaluating them on random subsamples (this could be viewed as a form of a mean-zero noise addition mechanism; we will clarify this in the next revision). Thus, our lower bound shows that all known (efficient) techniques for answering a super-linear number of queries in the i.i.d. setting provably fail when correlations are present in the data. As we mentioned at the end of Section 1.1.1, our work steers future work to either impose additional structural assumptions on the problem or introduce fundamentally new algorithmic techniques. Together with our simplified (and improved) positive results, we view this as telling a compelling story.
>
>
> **> This lower bound is not as surprising, since the concentrated queries assumption is extremely weak... When the dataset is given by i.i.d. samples, the random variable $q(D)$ is $1/\sqrt{n}$ subgaussian. Would a strengthening of the conditions to one of this form be enough to obtain an improved upper bound?**
>
> The weakness of the concentration assumptions is indeed a limitation of the framework (as our results highlight through the lower bound). We agree that it would be interesting to strengthen these assumptions or to focus on specific distribution structures that arise in practice. We hope our work provides a starting point for future work along these lines.
> Your suggestion about requiring concentration for every value of $\varepsilon$ is particularly interesting. While we do not answer this question, our attack and analysis offer some initial insight: due to the tradeoff between $\varepsilon$ and $\gamma_{\rm \scriptsize{Hoeffding}}$​ one can take $\varepsilon^{\*}=O(1/\sqrt(n))$ and proceed with the same construction and information-gathering rounds (since in that regime for any smaller values of $\varepsilon$, $\gamma_{\rm \scriptsize{Hoeffding}} >1$, thus our information gathering queries remain concentrated for any value of $\varepsilon$). This implies that data leakage—and therefore membership inference and even full reconstruction of the true sample —remains possible at the rate $\tilde{O}(1/{\varepsilon^{\*}}^2)$. What changes under the stricter requirement is that the final query that "overfits" the dataset would no longer be allowed, as it is not concentrated relative to every value of $\varepsilon$. We hope that such observations could provide useful intuition for future work along the lines of the stronger setting you suggest.
>
> **> The mechanism only slightly improves parameters over the original Bassily and Fruend algorithm.**
>
> It is true that, quantitatively, our upper bound only slightly improves the previous upper bound. But the analysis is substantially simpler, which we view as an important contribution of our work. We stress that while our analysis is simpler, it does *not* follow from a direct application of the generalization properties of DP to our setting, as is done in the i.i.d. case, and requires our use of the thought experiment of the "three worlds" described in Section 1.1.4.
>
> **> One striking gap is that throughout the paper, it seems that it is never formally stated what is supposed to mean that the lower bound rules out any mechanism that operated by adding "any noise"... it is assumed that the outputs are clipped...**
>
> The definition of "noise addition mechanisms" is stated in lines 99-101. We meant mechanisms that answer each query using the exact empirical average, after adding to it random mean-zero noise (independent of the previous noises). We will clarify this further in the body of the paper.
>
> Since queries are bounded in [0,1], any response outside the range of [-1,2] would already violate the accuracy requirement. In other words, if an answer falls outside of the interval [-1,2] then the mechanism fails and we (the attacker) win. Thus, in the proof of the lower bound we allowed ourselves to assume that the answers are always in the range [-1,2]. We agree that the terminology we used about this can be made more precise, and we will restate this in the next revision of our paper.

---

### Decision · Program_Chairs · 2025-09-17

**Decision:**

Accept (poster)

**Comment:**

This paper studies the problem of answering adaptive queries under the restriction that the queries concentrated around their means. The paper uses differential privacy to provide a simplified upper bound matching previous results by Bassily and Freund, as well as nearly tight query bounds for noise-additive mechanisms. Although all reviewers agree that the lower bound essentially closes the gap quantitatively, all reviewers also initially expressed concerns about the richness of noise-additive mechanisms. However, these concerns were addressed during the author-reviewer discussion phase.

I would encourage the authors to address reviewer concerns and to include expanded versions of the author response regarding noise-additive mechanisms, e.g., subsampling, to strengthen the potential impact of the work.